# Past Themes and Tracking Research Trends in Entrepreneurship: A Co-Word, Cites and Usage Count Analysis

**Luis Javier Cabeza Ramírez, Sandra M. Sánchez-Cañizares *** and **Fernando J. Fuentes-García**

Faculty of Law, Business and Economic Sciences, University of Cordoba, Puerta Nueva s/n, 14071 Cordoba, Spain; r62caral@uco.es (L.J.C.R.); fernando.fuentes@uco.es (F.J.F.-G.)

***** Correspondence: sandra.sanchez@uco.es; Tel.: +34-957-212-688

**Abstract:** This paper examines the evolution of research in Entrepreneurship published in Web of Science, a reference database. A bibliometric content analysis has been carried out as part of this investigation, allowing for a longitudinal study of the main research topics dealt with over time, ranging from classic topics such as its conception to more recent realities that include Social and Sustainable Entrepreneurship. This paper locates research trends by studying the evolution of citations and by incorporating use metrics. The results point to the existence of seven cognitive fronts that have marked the field's growth and conceptual evolution. Furthermore, evidence is presented that shows how innovation has historically been the thread that links all the core themes. The topics and trends detected contribute specially to advancing the current discussion on entrepreneurship and coordinating future research efforts.

**Keywords:** entrepreneurship; co-word analysis; research trend; bibliometric; science mapping; knowledge production

---

## 1. Introduction

Research into the structure of scientific fields is a topic that attracts the attention of the scientific community, as it helps to provide the basis for future advances and allows for the generation of new understanding built on the basis of pre-existing knowledge. Recently, [1] presented the results of a survey to researchers with experience in entrepreneurship focused on the most important thematic areas and those methods especially useful for advancing the understanding of entrepreneurship. The research presented complements this initiative by suggesting an alternative methodological proposal that focuses on the issues that have marked the evolution of entrepreneurship and helps to locate future research trends. For this purpose, a documentary corpus representative of the discipline is studied and bibliometric methods of content analysis (co-word) are applied to use the documents and their words to describe the evolution and current status of entrepreneurship.

The concept of entrepreneurship involves multiple realities, authors such as in [2] relate it to actions to encourage the creation of something new and valuable, despite this traditional approach aligned with the disciplinary conception of [3], linked to the innovative entrepreneur of [4] it can be excessively restrictive and leave out other related ways of understanding the entrepreneurship, for example, to self-employment or small businesses [5]. In recent years, progress is being made towards global approaches that include the role of the context in which entrepreneurship develops and emerge (entrepreneurial ecosystems) [6] and systems of entrepreneurship [7] that aim to unite the focus of the innovation system and entrepreneurship studies [6].

From the previous works of J. A. Shumpeter, it can be presumed that entrepreneurs are extraordinary people, who see opportunities where others do not. They assume risks and dedicate

their time and effort to exploit them. Some go so far as to engage in innovation. They usually draw economic and personal satisfaction from their activity. The concept of what constitutes entrepreneurship has been evolving. One of its unique characteristics when compared to other disciplines is its ability to adapt to the pace of society's advances and, at times, to act as the driving force behind such advances. Entrepreneurship's protagonists are able to detect social needs and transform such needs into opportunities [2,8]. This is what their survival depends on to a large extent, which is why entrepreneurs quickly absorb new realities they are confronted with. Meanwhile, classic definitions taken from management, economics and business studies have not taken on ideas relating to environmental protection. For instance, nowadays entrepreneurship has changed and is closer to new social paradigms and less dependent on traditional concepts. Sustainable entrepreneurship is beginning to attract the attention of researchers working in different areas ranging from social entrepreneurship, to environmental management [9–11], business innovation and market sustainability [12], integration of social and environmental aspects in medium-sized companies' products, processes and management [13], family entrepreneurship [14], globalisation and international entrepreneurship [15], spin-offs, regional entrepreneurship [16], technological entrepreneurship [17,18], rural [19], academic [20] and female entrepreneurship [2,21–24]. These topics can be the thousand faces of a single phenomenon.

However, what do we really know about entrepreneurship? What have been its core themes? How has it evolved into its current form? Where is research heading? Is entrepreneurship really a discipline? Even those authors who are most doubtful and critical with regard to the last of those questions point in this direction when confronted with the bibliometric evidence [25] (p. 519). It seems to be a logical conclusion then that the various bibliometric analyses have been strengthening and revitalising efforts to make theoretical advances by showing the direction research work is following.

A general analysis of the bibliometric studies to date shows that those papers which take an approach to the entrepreneurship as a global discipline (Appendix A Table A1) start to appear towards the end of the 90s [26,27] and serve as a reference for those other documents that deal with specific areas, which emerge when the concept of entrepreneurship begins to expand around the year 2011 (Appendix A Table A2) [14,15]. According to this idea, few works about bibliometrics applied to entrepreneurship have managed to get published, since more than 3000 documents related to entrepreneurship have been published only in Web of Science (WoS) and Scopus since 2010 (Appendix A Figures A2 and A3).

The article presented here is designed to bridge this gap by means of a content analysis, rarely performed until now through a co-word analysis. Moreover, this paper introduces something new, namely two significant improvements in tendency analysis: it presents the citations corresponding to a representative document sample at two different moments in time in order to reflect their evolution. It also employs in the discipline metrics linked to information use. Thus, a more realistic approximation to what researchers believe to be most influential can be obtained. The main objective of this paper is to define the intellectual structure of the entrepreneurship field, to understand how its main thematic lines have evolved and eventually led to the current concept of entrepreneurship, as well as to show which direction research is taking at the moment. A content analysis was carried out to that end [28] on a representative sample of entrepreneurship-related documents, and research trends were tracked by observing the evolution of citations from the sample over two years and a half of exposure. In addition, new metrics: usage count since 2013 and usage count in the last 180 days were used.

The results outline different cognitive fronts that have marked the growth of the discipline, and show how the close and complicated relationship between entrepreneurship and innovation [29] acts as a vertebral axis that connects both concepts. This issue should continue to be explored, taking into account the new concept of what constitutes an entrepreneur. This study provides a significant contribution to our theoretical understanding of the phenomenon. As [30] argued, if the field wants to progress, it needs to make advances in the search for entrepreneurship's own theories that cannot be explained from the point of view of other disciplines. Understanding the thematic evolution and

research trends will allow interested scientists to generate new research agendas and focus their efforts on those aspects which need closer attention in order to obtain answers that can satisfy new social demands.

The rest of the study is divided into four sections: a brief literature revision which contextualises the article, explaining what bibliometrics has contributed to trend analyses, followed by the main methodological considerations of the study; then, the presentation of the results obtained and, to conclude, a discussion of those results and the main conclusions.

## 2. Literature Review

Entrepreneurship is considered to be an engine of economic development. It generates growth and serves as a vehicle for innovation and change [31]. It has been widely studied in different fields and from different angles. Entrepreneurship is a phenomenon with a wide political and institutional recognition. Its potential benefits for society have led to the implementation of policies specifically designed to promote entrepreneurship, and most governments of the developed world dedicate large sums of money to cultivate it [32,33]. Entrepreneurs are highly regarded and few researchers have questioned their significance when it comes the impact they have on the creation of jobs and opportunities, on economic growth, on the promotion of an inclusive society, on wealth creation and national levels of competitivity and productivity [34–36].

Despite entrepreneurship's unquestionable popularity, which has led to an increasing number of researchers interested in deepening in their understanding of the phenomenon, there has always existed a series of questions which have caused profound debate. They are concerned with the way research is performed, and with entrepreneurship's configuration as an academic discipline as well as with the definition of what constitutes an entrepreneur [37]. In that sense, one of the most recognised reviews of entrepreneurship [38] focused on two aspects considered fundamental for generating progress in research: the need to improve "research design specifications" and the importance of compiling all accumulated knowledge when attempting to generate advances, because "as a body of literature develops, it is useful to stop occasionally, take inventory of the work that has been done, and identify new directions and challenges for the future" (p. 139).

The discipline is in full development although it is still young, highly fragmented and unstructured. One paper [39] "considers entrepreneurship research as a 'melting pot' of concepts and theories from many different disciplines" (p. 46). The literature about entrepreneurship has multiplied at an exponential rate. In the 1990s, there was an average of 350 documents per year directly or indirectly related to entrepreneurship in Web of Science. Currently, the Core Collection of this database and Scoups has reached historical peaks over 6000 documents per year (year 2017; search = topic = entepr\*, Appendix A Figures A1 and A2), thus reaching an unprecedented number of works and involving more and more research sectors and fields. Under these circumstances, tracking research trends has become an absolute necessity for entrepreneurship researchers and reached a certain level of complexity. Traditionally, qualitative approaches were used similar to those typical of structured bibliographical reviews [38,40] or systematic literature reviews [41], which have gradually led to more objective and adequate methods such as meta-analyses [42,43] or bibliometric analyses [29,44,45].

Research trends are the combined scientific ideas that drive research into a topic, area, field or discipline in a certain direction. Trends absorb the propensity or direction of scientists' work and, consequently, bibliometric studies using citations, references or words as a measure for impact become instruments especially appropriate for identifying such trends. Bibliometric analyses tend to follow a systematic review process when choosing and analysing documents and are based on the public validation of research materials by the discipline main actors. This kind of method is not new. It became more common with the emergence and more generalised use of online databases, and after bibliometric software was developed that facilitated the treatment of large quantities of bibliographic data [46]. The application of bibliometrics in the context of entrepreneurship is gaining more and more significance and allows its researchers to progress in its understanding. In the case of the definition of

what actually constitutes an entrepreneur or the discipline as a whole, bibliometrics and its researchers point us towards the writings of J.A. Schumpeter and the conception of the discipline by [3] as the most influential works because they are, in objective terms, the most cited and referenced.

Bibliometric investigations are not free from limitations and have led to a profound reflection in the academic community [47]. The information they provide must be handled with care and guided by a solid theoretical understanding of the actual characteristics of the discipline or area where they are applied. The latest novelties in bibliometrics have been the introduction of new kinds of metrics relating to scientific document consumption and social development [48–52]. The so-called user metrics and Altmetrics complete the information that the usual indicators transmit and allow for new data to be obtained pertaining to new research habits such as the downloading of documents [53] or use of social networks. Bibliometric research in entrepreneurship is not as highly developed as it is in other areas such as, for example, Information Science, where its use became widespread much earlier. It was not until the 1990s when this type of tool was introduced in entrepreneurship [54,55] in papers centred around small businesses. Consequently, there is little accumulated experience and there are still few analyses concerned with annual document production. Research papers that have compiled the main bibliometric studies in entrepreneurship as part of their analysis are few [39,56,57]. Among those few, two types of approach can be distinguished: those that use bibliometrics to study the discipline in general (Appendix A Table A1) and those that focus on specific areas pertaining to their field of expertise (Appendix A Table A2). Moreover, there is a significant deficit in works centred on the actual content of the documents under investigation (co-word analysis) or the references they contain (bibliographic coupling), particularly in Management [46] and the general analyses of entrepreneurship [56].

## 3. Materials and Methods

The methodology applied corresponds to the approach detailed in Figure 1. The majority of bibliometric studies in entrepreneurship indicate their static nature as a limitation: "the research sketches a static portrait, whereas structuration of the field is dynamic" [26] (p. 303), "a third limitation of this study is that it is static. This paper measured the impact of scholars and institutions on entrepreneurship research at one moment in time" [27] (p. 94), "our ACA shows a static snapshot of entrepreneurship" [58] (p. 427), "the analysis although involving a rather long time span is quite static" [57] (p. 52). Citations, however, are in progress and, consequently, bibliometric analyses that look at the same citations at two different moments in time become more dynamic. In this case, two years and a half elapsed between one snapshot and the other, which left enough time for citations to accumulate. Furthermore, new research methods tend to require the most influential articles to be downloaded; thus, two more indicators can be added: usage counts starting in 2013 and counts from the last 180 days prior to the study. That way, research trends can be traced by detecting those articles that have shown the greatest evolution in terms of their increase in citations obtained and those that have been used/downloaded the most.

The representative entrepreneurship-related document sample is obtained through the H-Classics methodology proposed by [59]. This denomination (Classics) is not to be understood literally, it refers to works with citation rates above the H-Index of a certain area in a certain moment of time. To apply this methodology, a search was performed using the root "entrepr*" in a single database, thus avoiding different citation patterns for the same document according to different databases. In this case, Web of Science was chosen although the possibility of contrasting the results in the future with other reference databases such as Scopus or Google Scholar is open. The decision to use a single search term was controversial, however, it is a previously used strategy [44,60] that according to [44] avoids biasing the results towards areas particularly familiar for researchers ("small firms", "small enterprises", "entry firms", etc.). Although this means a limitation, the question has not yet been solved in the literature. Next, according to the multidisciplinary nature of the entrepreneurship [29] the results obtained were filtered according to the different areas of knowledge through the search sequence

described in Table A3 of the Appendix A. Finally, the H-Index of the whole was determined (201 in June 2016) and those documents with a number of citations above this figure were recovered. The final sample consisted of 205, four documents above the H-Index added to prevent possible eliminations. A full characterization can be found in [61]. Some basic data about the sample are given in the Appendix A Figure A1 and Tables A4–A9.

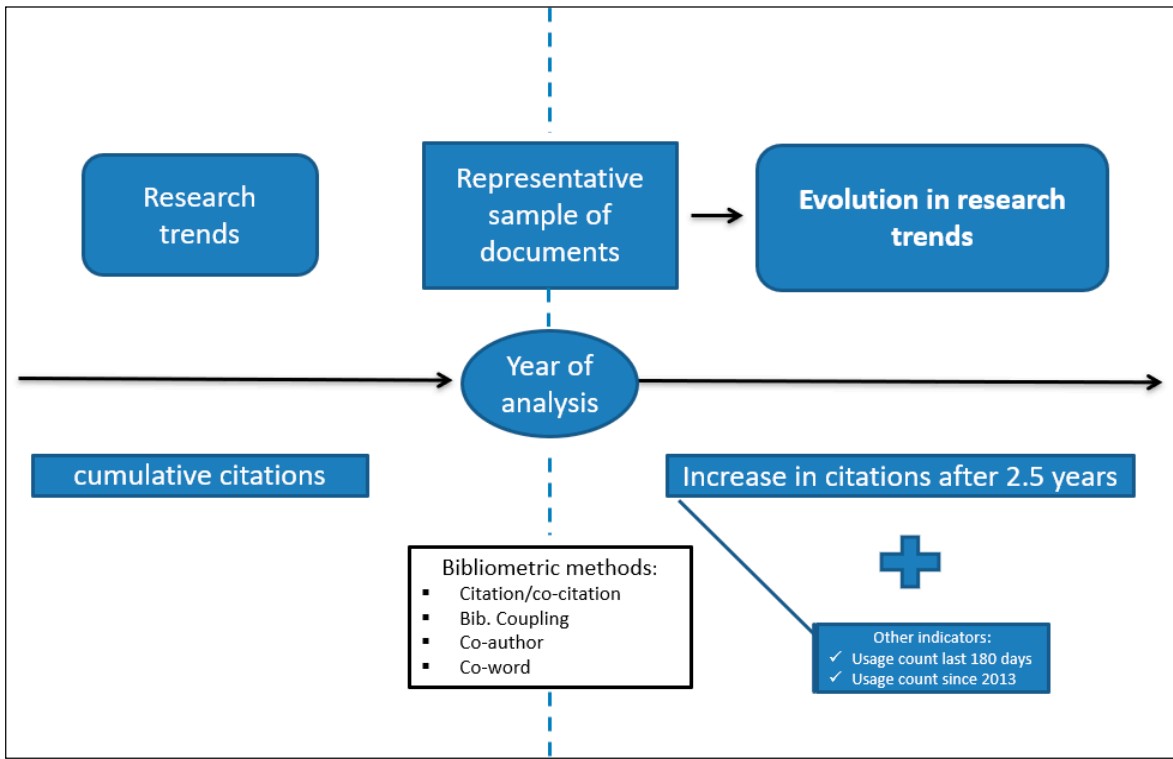

**Figure 1.** Model to track research trends using bibliometric methods.

Afterwards, a content analysis of the documents was carried out using a co-word analysis [28] in order to trace the evolution of the core themes. Co-words analyses are based on the assumption that the keywords of a document or scientific paper provide an adequate description of the content, as two words will co-occur in documents that address similar themes and there will be links between them [62–64]. Many co-occurrences for the same word or set of words give rise to a "strategic alliance" between documents associated with a specific research topic. The 205 most relevant documents, obtained through the application of the H-Classics method [59], have been used as the starting point for this analysis. The works and authors that comprise the sample (Supplementary Materials Table S1) have been called "classics" and the references they contain are part of the knowledge base. These works correspond to early researchers and documents that have become visible via these classics (applying the terminology of [45,60]). The documents were then entered into SciMAT (University of Granada, Granada, Spain) (a software tool developed by [65] to perform science mapping analyses) and the different steps of the science mapping workflow were performed as follows: pre-processing and normalization of the document sample, addition of keywords (following the recommendations of [28,66,67], extraction and normalization of the bibliometric network (the network of associated words was normalised using the equivalence index following [63,68]) and mapping (the simple centres algorithm was used to build the map [60,64,69]).

The thematic networks generated from the workflow represent research topics according to the co-occurrence of keywords for each time period. With the list of terms included in the classics, graphs are created where the nodes are the keywords and the links between them are their relationships. When keywords appear in the same documents, links are made between the nodes. By adding weight to

these links, the importance of the relationship within the set of documents that compose the research is represented. If the relationship is quantified, the matrix of associations of co-occurrences is constructed (keyword x keyword). This is a symmetric quadratic adjacency matrix in which each element represents the association between descriptors. When the simple centre cluster algorithm is applied to the matrix normalised by the equivalence index, the words are grouped into themes and the thematic network is built (Figure 2a). Each network is labelled using the most significant keyword contained in the network (usually the most central keyword of the cluster) and a set of themes is obtained for each period.

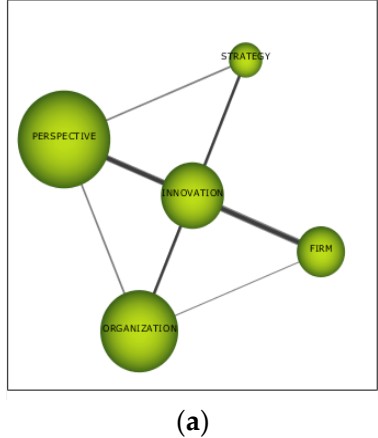

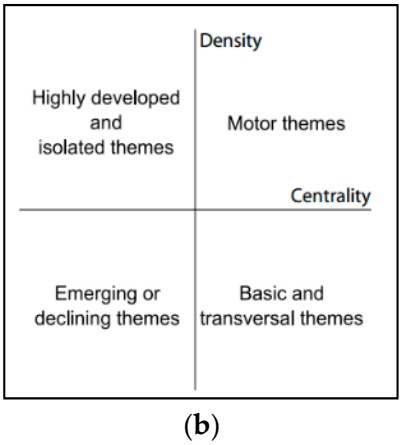

(**a**)  (**b**)

**Figure 2.** Example of (**a**) thematic network and (**b**) Strategic Diagram Source: [55].

From the set of thematic networks, strategic diagrams are created to reduce the space of words comprising the set. By means of an aggregation process using Callon's centrality and density measures and their respective ranges [63], each aggregate is placed on a Cartesian axis where X and Y are the centrality and density that define four regions (Figure 2b), thus making it easier to understand the information they provide.

- Quadrant 1 corresponds to the core of the community. These are aggregates with a high degree of development and integration. This quadrant comprises subjects with strong centrality and high density and therefore contains the motor themes of the field. According to [70], lk a motor theme is derived from well-established knowledge (high density), and has implications to new topics (high centrality)
- Quadrant 2 comprises basic and transversal themes which are highly developed aggregates with high density and low centrality. They may be motor themes that have become isolated over time owing to a fade in interest.
- Quadrant 3 includes peripheral themes that are well developed internally but isolated from other themes and play a marginal role in the development of the scientific field.
- Quadrant 4 corresponds to emerging or declining themes with low centrality and density that are well connected but underdeveloped.

Finally, the inclusion index of [71] (Figure 3a) and the evolution in the number of common elements (words) between consecutive periods according to the degree of overlap or stability index between periods (Figure 3b) are used to determine how the different themes detected during the selected periods have evolved.

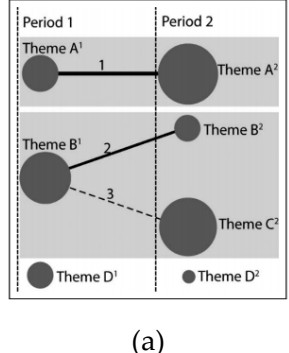　　　　　　　　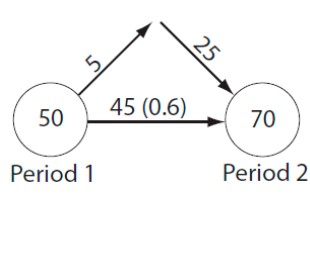

　　　　　　(a)　　　　　　　　　　　　　　　　　　　　(b)

**Figure 3.** Example of (**a**) evolution areas in the number of common elements; (**b**) stability between periods, source: [55].

## 4. Results

In what follows, the method described in [61] is applied to the documents and characterisations previously mentioned in order to visualise key concepts that are representative of the research in the classic publications of entrepreneurship literature. The results will be enriched with bibliometric indicators to assess the impact, quality and performance of these themes (number of documents, number of citations and average number of citations).

### 4.1. Conceptual Evolution of the Discipline Through Its Classics: A Co-Word Analysis

The sample has been divided into four time periods according to the number of documents per year of publication and citations received (the periods of analysis were selected to ensure a balance in terms of number of documents). The first period, which we call "origins," contains 45 documents and corresponds to the oldest documents (1968–1995). The second and third periods (1996–2000 and 2001–2005), referred to as development 1 and 2, contain 54 and 86 documents, respectively, and cover the time span that has had the greatest impact within the community. The fourth period (2006–2011) includes only 20 more recent publications that are in the process of consolidation. This period includes documents that have had less exposure to other authors; hence the smaller number of documents and citations.

### 4.1.1. First Period 1968–1995: Origins

Within the four periods selected, this period is the longest and least homogeneous in terms of the distribution of documents. It covers works mainly dating from 1988 onwards. The period includes 45 classics with the oldest publication dates. As can be observed in the strategic diagram representing the period (Figure 4 with total number of citations), there are nine themes, which are subdivided as follows:

- Motor themes (Quadrant 1): Performance, Individual-Trait, Corporate-Entrepreneurship and Motivation
- Basic themes (Quadrant 2): Entrepreneurial-Firm
- Highly developed and isolated themes (Quadrant 3): Success
- Emerging themes (Quadrant 4): Organizational-Structure, Market Imperfection and Model

These themes form a complex and highly specialised organizational structure of the network. The bibliometric indicators for performance and impact associated with research topics are shown in Table 1 Period 1 Origins. According to these metrics, the most relevant themes for the scientific community have been Performance (motor), Entrepreneurial-Firms (basic) and Organizational-Structure (emerging). The themes Success (isolated) and Market-Imperfection (emerging) have been the least cited overall, that is, these themes had the least impact in subsequent years (it is worth noting that we

are dealing with classic documents that already have the highest citation rates within the discipline). It should be clarified, according to the nature of the co-words analysis that extracts the co-occurrences of terms in the different periods of time, that the number of total documents does not necessarily have to be the same than the number of documents in which the terms most significant detected for each period of time co-occur. For example, this first period contains 45 documents but only the words detected in 34 of them co-occur, and the same happens with the other three periods of time.

**Table 1.** Summary of bibliometric indicators.

| 1st Period. Origins (1968–1995) | | | |
|---|---|---|---|
| **Research Topics** | **Number of Documents** | **Average Number of Citations** | **Sum of Citations** |
| Performance | 4 | 651.25 | 2605 |
| Motivation | 5 | 360.40 | 1802 |
| Entrepreneurial-Firms | 7 | 386.71 | 2707 |
| Individual-Trait | 4 | 441.50 | 1766 |
| Corporate-Entrepreneurship | 4 | 327.75 | 1311 |
| Organizational-Structure | 4 | 577.25 | 2309 |
| Market-Imperfection | 2 | 385.00 | 770 |
| Success | 2 | 393.00 | 786 |
| Model | 2 | 509.50 | 1019 |

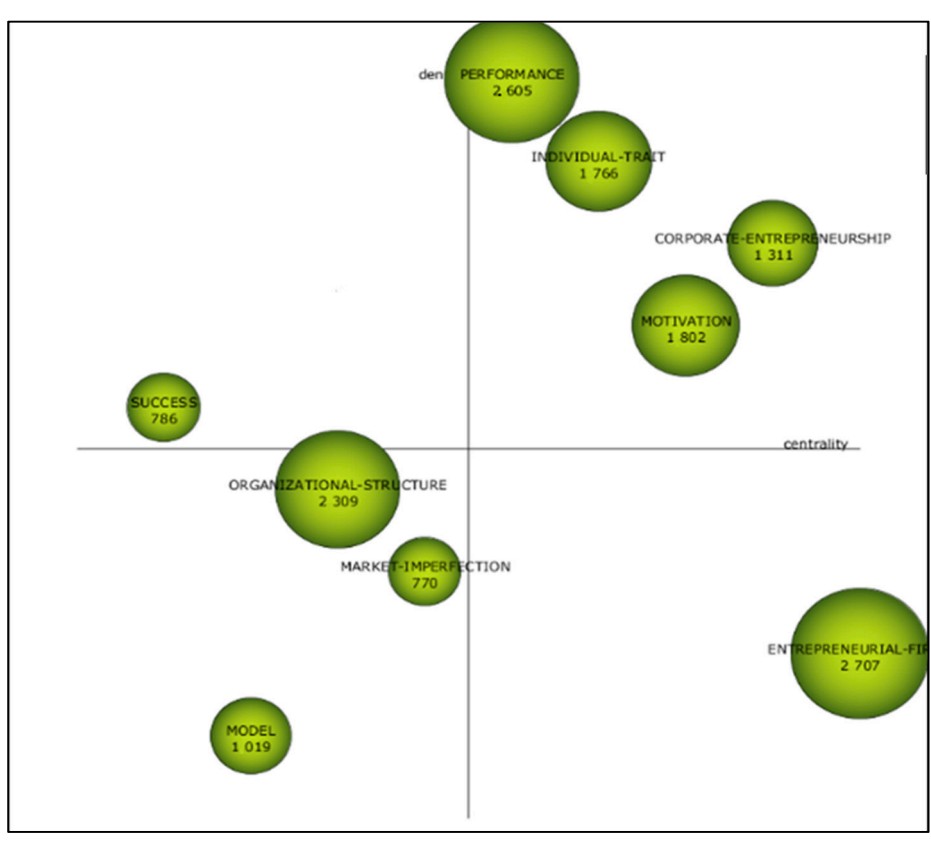

**Figure 4.** Strategic diagram (1968–1995) with sum of citations.

### 4.1.2. Second Period 1996–2000: Development 1

This period shows a more homogeneous temporal distribution than the previous one. It comprises a total of 54 classics over a period of five years and corresponds to a stage in which literature on entrepreneurship became more established. Along with the following period (Development 2), it is the period with the largest number of works as well as the highest citation rates. It comprises ten research topics and, like the previous period, displays a complex and rich network with a high degree

of specialisation. The strategic diagram (Figure 5 by number of documents) shows its main themes (motor): Management, Perspective and Industry. The overall configuration of the period is subdivided as follows:

- Three basic themes (Quadrant 2): Market, Entrepreneurship and Performance
- Three highly developed and isolated themes (Quadrant 3): Orientation, Competition and Self-Employment
- One emerging theme (Quadrant 4): Managers

The associated indicators for bibliometric performance (Table 2 Period 2) show how the basic theme, Market, has the greatest impact. This theme includes the first article by cites of the H-Classics [72], followed by the motor theme, Perspective, and another basic theme, Entrepreneurship. Finally, very developed but isolated themes (Orientation, Competition and Self-Employment) rank last. It is also worth noting the evolution in this period of the central thematic core associated with Performance, which was maintained over the period 1968–1995 and changed from motor theme to basic theme.

**Table 2.** Summary of bibliometric indicators.

| 2nd Period. Development 1 (1996–2000) | | | |
|---|---|---|---|
| **Research Topics** | **Number of Documents** | **Average Number of Citations** | **Sum of Citations** |
| Management | 4 | 559.50 | 2238 |
| Industry | 4 | 456.50 | 1826 |
| Perspective | 8 | 402.00 | 3216 |
| Entrepreneurship | 8 | 361.75 | 2894 |
| Performance | 6 | 426.33 | 2558 |
| Market | 5 | 913.20 | 4566 |
| Competition | 2 | 252.00 | 504 |
| Orientation | 2 | 233.50 | 467 |
| Self-Employment | 2 | 447.00 | 894 |
| Managers | 2 | 1219.00 | 2438 |

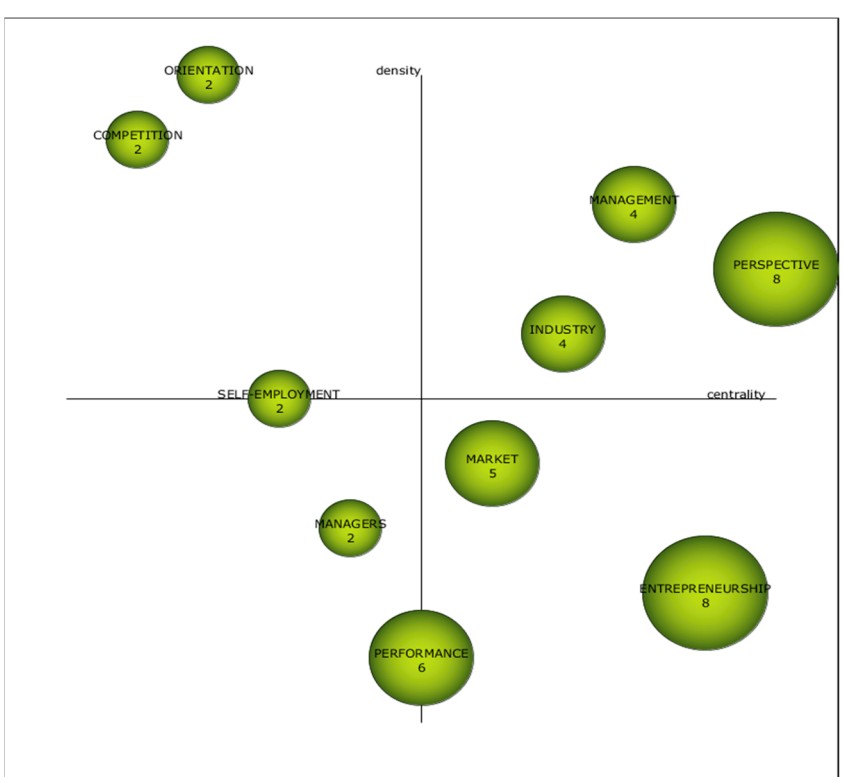

**Figure 5.** Strategic diagram (1996–2000) by number of documents.

### 4.1.3. Third Period 2001–2005: Development 2

This period includes the largest set of documents (86). There is a higher concentration of classics in the period, which is as homogeneous as the previous one. It coincides with the most representative stage of creation in the field. The strategic diagram for the period (Figure 6 by average citations) represents nine research topics. The first quadrant includes four motor themes: Discovery, Evolution, Competitive-Advantage and Innovation. The second theme, Strategy, is a basic theme. The third, Joint-Ventures, is a highly developed and isolated theme, while the fourth includes two themes that are disappearing, Market and Strategic-Alliances, and one emerging theme, Model.

Regarding the distribution in terms of relevance and impact, Table 3 Period 3 shows that the motor themes Competitive-Advantage and Innovation are the most cited and account for the largest number of citations. In contrast, Joint-Ventures, Strategic-Alliances and Market have the least impact and are the least cited. Moreover, Market has evolved since the previous period and is disappearing.

**Table 3.** Summary of bibliometric indicators.

| 3rd Period. Development 2 (2001–2005) | | | |
|---|---|---|---|
| **Research Topics** | **Number of Documents** | **Average Number of Citations** | **Sum of Citations** |
| Evolution | 11 | 333.36 | 3667 |
| Competitive-Advantage | 15 | 354.67 | 5320 |
| Innovation | 18 | 343.17 | 6177 |
| Strategy | 11 | 387.09 | 4258 |
| Discovery | 4 | 335.00 | 1340 |
| Model | 6 | 245.83 | 1475 |
| Joint-Ventures | 3 | 323.00 | 969 |
| Market | 3 | 389.33 | 1168 |
| Strategic-Alliances | 3 | 319.00 | 957 |

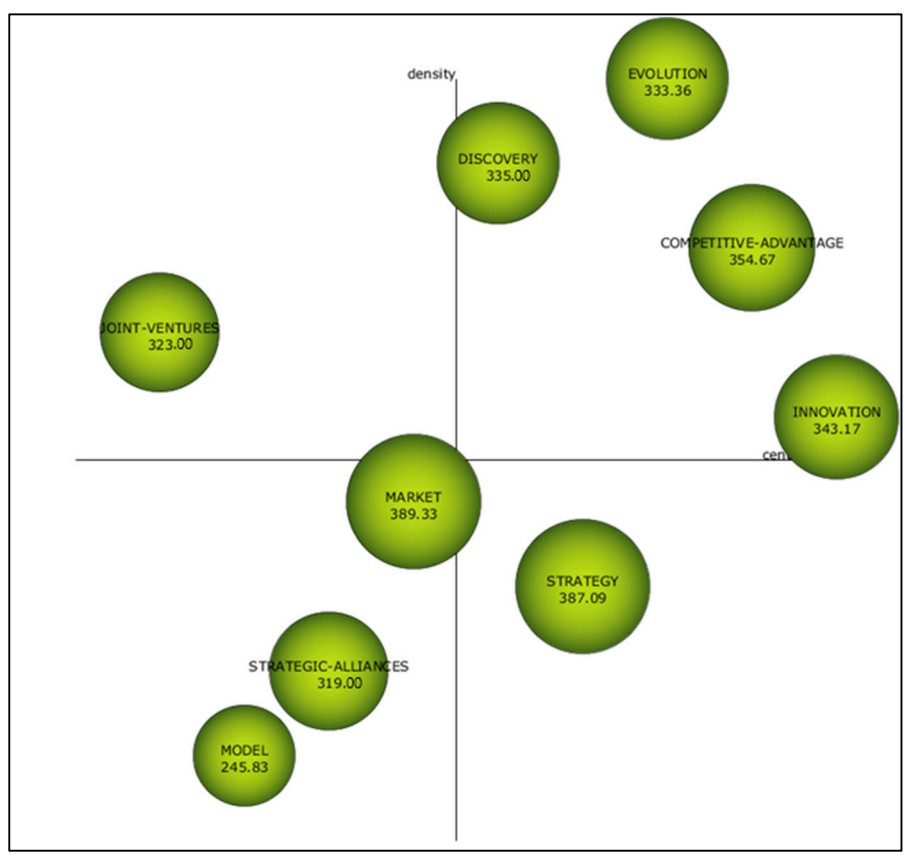

**Figure 6.** Strategic diagram (2001–2005) by average number of citations.

### 4.1.4. Four Period 2006–2011: Consolidation

The last of the periods, despite being one year longer than the previous two (6), comprises just 20 classic documents. This is the period in which work in entrepreneurship has become widespread, but most of the research has not yet managed to be among the most cited or requires more exposure time to accumulate citations. The number of documents in the strategic diagram attest to this fact (Figure 7), as it maps a network still in the process of becoming structured (it mainly occupies the second bisection).

The period has three motor themes that reflect a greater interest in field studies and studies on Transformation, while Innovation continues to be one of the main focuses of knowledge. Quadrants 2 and 3, which refer to basic and isolated themes, are very close to Quadrant 4, which captures emerging themes. These quadrants include the rest of the key themes for the period: Social-Value, Absorptive Capacity, Model and Performance. Of the seven themes that represent the period, two were already prominent in previous periods: Model, which remains an emerging theme as in the period 2001–2005, and Performance, the basic theme during the period 1968–1995, re-emerges. The performance measures shown in Table 4 (period 4) indicate that the themes developed in this period have had less impact (always measured in relative terms as they are H-Classics). The theme Innovation is more prominent than the rest, while the theme Model shows the lowest citation rates.

**Table 4.** Summary of bibliometric indicators.

| 4th Period. Consolidation (2006–2011) | | | |
|---|---|---|---|
| **Research Topics** | **Number of Documents** | **Average Number of Citations** | **Sum of Citations** |
| Innovation | 3 | 581 | 1743 |
| Transformation | 4 | 297.5 | 1190 |
| Field | 4 | 225.25 | 901 |
| Performance | 4 | 295.5 | 1181 |
| Absorptive-Capacity | 2 | 287.5 | 575 |
| Social-Value | 2 | 346.5 | 693 |
| Model | 2 | 211.5 | 423 |

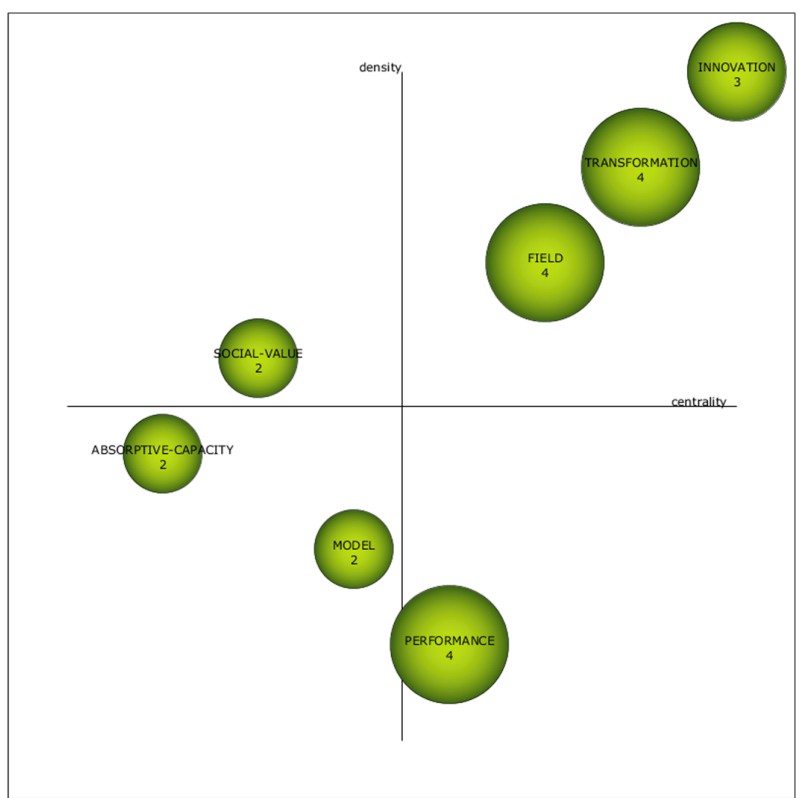

**Figure 7.** Strategic diagram (2006–2011) by number of documents.

4.1.5. Evolutionary Map

The set of descriptors or key terms of the documents is not constant across the different periods (Figure 8). Terminology changes, different keywords are used to map the content of the classics, new words emerge and others disappear. For example, the terms Access-to-Capital, Block model and Business-Assistance, which correspond to the first period, appear only in that period, while Innovation or Performance are present in most of the periods studied.

As can be seen in Figure 8, the map is very dense. Some themes belong to more than one area simultaneously, while others are isolated or have lacked the continuity necessary to play a prominent role in the classics (Success, Organizational-Structure, Orientation, Competition, Joint-Venture and Social-Value).

- First period 1968–1995: This period contains 232 keywords, 84 of which re-appear in the following period (1996–2000). The remaining 148 keywords do not appear in the following period. The similarity index between the first and second periods is 0.1.
- Second period 1996–2000: This period contains 310 words, 226 of which are new. In the following period, 128 of these words re-appear and 182 are not used again. The similarity index between the second and third period is 0.13.
- Third period 2001–2005: With 409 words, this period has the largest number of words. A total of 281 new words have been incorporated. In the following period, 63 of the words remain and 346 disappear. The similarity index between the third and fourth period is 0.07.
- Fourth period 2006–2011: Given that there are few documents in this period, there is also a smaller number of keywords. Of the 139 keywords in this period, 76 are new additions.

The longitudinal analysis offers a global vision of the evolution through the four periods. The bibliometric software (SciMAT) offers multiple possibilities, the word network is normalised using the equivalence index and the global map is constructed with the algorithm of simple centres [69,73,74]. The parameters used are described in Table A10 of the Appendix A. The size of the spheres is proportional to the number of documents associated with the central keyword of the subject. The solid lines of each link mean that the linked topics share the same name, that is, these topics have been tagged with the same keyword (the most central word in the formed centre). Dashed lines mean that the themes share elements that are not the central core. The thickness of the lines between topics is proportional to the inclusion index between the linked topics. The interpretation of the map is achieved by following the path of each of the subjects, as shown in Figure 9 (the first three paths are described) and observing whether the central nuclei labelled, are motor themes, basic and transversal, highly developed or isolated or emerging or declining according to Figure 2b (methodology). The main cognitive lines formed are assigned a denomination as a summary of their evolution.

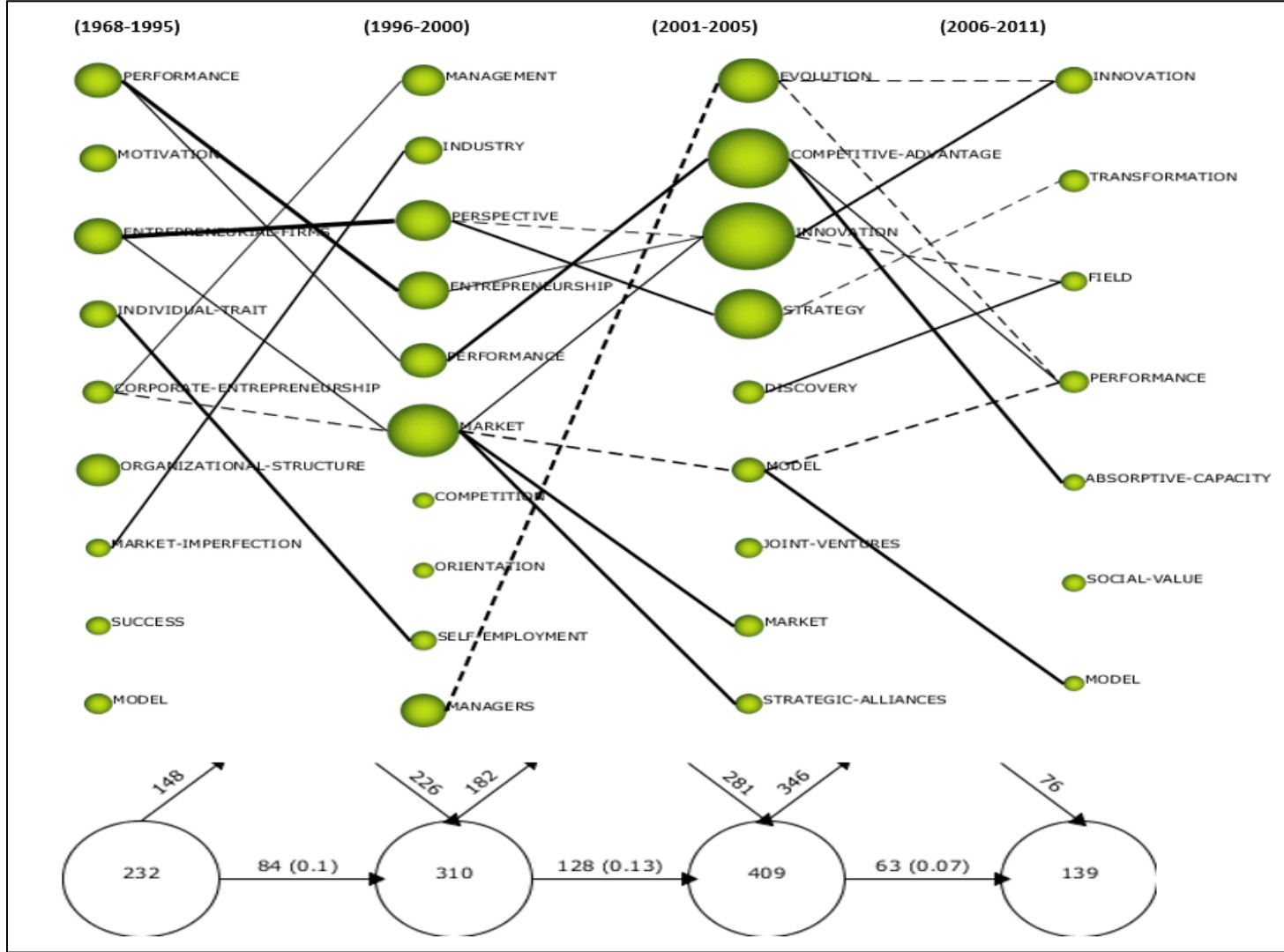

**Figure 8.** Evolutionary Map and Stability index between periods.

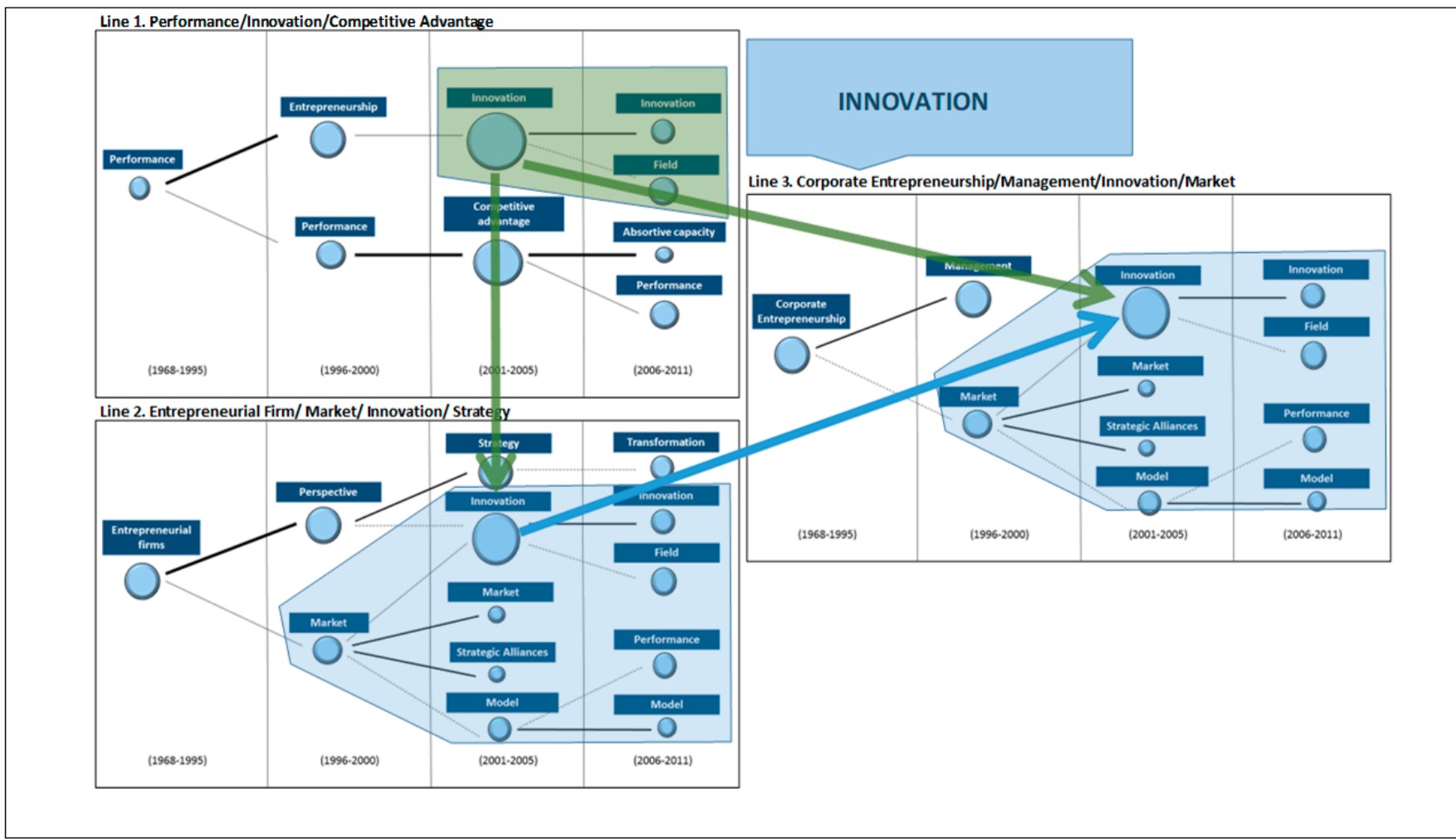

**Figure 9.** Innovation as a link between the lines with the greatest impact in the discipline.

Table 5 shows the most important relationships within each thematic area or cognitive line, with two measures of quality (mean/sum of citations), as well as the number of documents that represent them. Each line has been assigned a label that includes the main relationships that comprise it.

**Table 5.** Measures of quality for each cognitive line.

| Cognitive Lines | Number of Documents * | Average of Citations | Sum of Citations |
|---|---|---|---|
| Performance/Innovation/Competitive Advantage (Line 1) | 64 | 342.95 | 21,949 |
| Entrepreneurial Firm/Market/Innovation/Strategy (Line 2) | 69 | 400.74 | 27,651 |
| Corporate Entrepr./Management/Innovation/Market (Line 3) | 56 | 375.66 | 21,037 |
| Individual Trait and Self-Employment (Line 4) | 6 | 443.33 | 2660 |
| Market Imperfections and Industry (Line 5) | 6 | 432.67 | 2596 |
| Managers/Evolution/Performance/Innovation (Line 6) | 20 | 438.65 | 8773 |
| Discovery/Field (Line 7) | 8 | 280.13 | 2241 |

* The total sum of documents is higher than the total sample because the same document can be included in several lines.

From the thematic composition it can be observed that:

- Line 1 is the most solid, since it is composed of motor and basic themes in all the periods studied.
- The following two thematic areas (lines 2 and 3) are the most important and are currently in the process of being developed. These lines exhibit the most ideal evolutionary behaviour and are expanding through the motor and basic themes, which are the origin of new emerging themes.
- There are two peripheral or specific cognitive lines (lines 4 and 5). These are research areas that have sparked the interest of the scientific community in specific periods of time but show signs of exhaustion owing to a lack of continuity.
- The last two lines (6,7) are in its peak and reach the last period in the form of motor themes (Innovation and Field). They arise from emerging themes (Managers) and motor (Discovery) and enjoy sustained growth that is consolidated. These lines may be the origin of new thematic areas in future.
- The development of entrepreneurship as a scientific discipline, reflected in the thematic areas detected in the classics, shows strong cohesion, as most of the identified themes weave a thread that runs through the different periods into which the study is divided.
- The first three lines are the ones with the greatest impact (Table 2 shaded in blue). These lines are grounded in a theme that clearly draws from Economics, Management and Business. These lines of development are intertwined especially in the last two periods (2001–2005 and 2006–2011), with Innovation acting as a link between them and as a real catalyst for the advancement of the discipline (Figure 9).
- Certain themes are not associated with any particular line (Motivation, Organizational-Structure, Success, Competition, Orientation, Joint-ventures, Social-Value). These are isolated themes that had an impact in a given period but which are difficult to insert in a particular thematic area, either because they are emerging themes (i.e., Social-Value) or because they are linked to too many thematic areas and are too general (i.e., Success, Orientation, Joint-Ventures).
- The evolution in the number of documents (size of the spheres) is homogeneous across the four periods, with some exceptions: Market, Competitive-Advantage, Innovation and Strategy, all of

which comprise a similar number of documents. The number of highly cited works addressing these themes has grown, thus indicating that they are of increasing interest in the field.

- Overall, line 2 (Firm, Market, Innovation and Strategy) shows the best impact indicators.

### 4.1.6. Research Trends: Evolution of Citations and Usage Counts

The previous section outlined the thematic evolution and provided a representative entrepreneurship-related document sample with high citation rates. This section attempts to visualise the variation in citation Figures after two and a half years of exposure. It is worth noting that the behaviour of citations is not exclusively motivated by the desire to recognise the intellectual or cognitive impact of scientists working in the same area. There are other factors that play a role in the decision to cite [75].

Firstly, it can be observed that increases in citation figures oscillate between those documents that barely reach a 20% [76,77], both of them are "old" documents (1995, 1998), whose connection to entrepreneurship is rather indirect, their main focus being Finance, and those [78,79] reaching up to 181% and 218%. These are more recent documents with shorter exposure time and directly related to Entrepreneurship, Business Model and Entrepreneurship Education. The average increase in citations per article was 253, i.e., a 66% (these are documents that already had high citation rates).

Table 6 shows a ranking of the 10 best-positioned works. The first position is occupied by a review of business models [78], which focuses the attention on different conceptualisations and on how business models attempt to explain value creation and capture. The ranking reflects a balance between usual topics related to Entrepreneurship such as Entrepreneurial Orientation and Intention [80–82] and, as a result of a significant increase in citations, the emergence in leading positions of documents centred on specific topics such as Social or Institutional Entrepreneurship [83–85]. It is also relevant to observe the year of publication of the first ten ranks. All except [80] were published from 2005 onwards. Among those ranked highest are some articles with the least exposure time of the sample, such as [78], which had accumulated 256 citations in 2016, and 815 citations in 2018 after only 8 years of exposure. This contrasts with other documents such as [86] with 240 citations in 2016 and 358 in 2018, after 51 years of exposure. This reflects how the ranking visualises documents which mark tendencies and capture the interest of the community.

**Table 6.** Top 10 Documents with higher cites increase.

|  | Document ID | Year | Cites 2016 | Cites 2018 | Increase | % | Topic |
|---|---|---|---|---|---|---|---|
| 1 | [78] | 2011 | 256 | 815 | 559 | 218% | Business Model |
| 2 | [79] | 2005 | 210 | 589 | 379 | 181% | Entrepreneurship Education |
| 3 | [87] | 2005 | 206 | 550 | 344 | 167% | Knowledge Economy |
| 4 | [88] | 2011 | 230 | 612 | 382 | 166% | Institution |
| 5 | [83] | 2006 | 339 | 883 | 544 | 161% | Social Entrepreneurship |
| 6 | [82] | 2009 | 258 | 669 | 411 | 159% | Entrepreneurial Orientation |
| 7 | [84] | 2006 | 354 | 868 | 514 | 145% | Social Entrepreneurship |
| 8 | [85] | 2009 | 244 | 572 | 328 | 134% | Institutional Entrepreneurship |
| 9 | [80] | 2000 | 475 | 1112 | 637 | 134% | Entrepreneurial Intention |
| 10 | [81] | 2005 | 270 | 621 | 351 | 130% | Entrepreneurial Intention |

In addition to that, Thomas Reuters's usage count offers information relating to the number of times that a document was accessed in its entirety or a copy was downloaded. This is an alternative measure, which makes sense, as researchers download and keep a copy of those documents that are most necessary for their work. Just like citations, this measure also suffers some limitations, as only those documents that can be downloaded completely can have an impact on this indicator. It also depends in large part on the access to subscriptions that researcher's academic institutions provide. In spite of this, this measure offers valuable and complementary information that can help detect trends. Table 7 shows a ranking of the ten most used/downloaded documents:

**Table 7.** Top 10 Documents by Usage Count in the last 180 days and since 2013.

| Document ID | Year | Cites 2016 | Cites 2018 | Usage Count 180d | Rank Usage Count 180d | Usage Count Since 2013 | Rank Usage Count 2013 | Topic |
|---|---|---|---|---|---|---|---|---|
| [89] | 2007 | 1124 | 2498 | 124 | 1 | 1548 | 1 | Sustainable Enterprise Performance |
| [78] | 2011 | 256 | 815 | 106 | 2 | 978 | 2 | Business Model |
| [90] | 2001 | 785 | 1389 | 91 | 3 | 972 | 3 | E-business |
| [80] | 2000 | 475 | 1112 | 84 | 4 | 887 | 5 | Entrepreneurial Intention |
| [91] | 2000 | 1020 | 1573 | 65 | 5 | 599 | 8 | Entrepreneurial Opportunities |
| [3] | 2000 | 2090 | 3741 | 64 | 6 | 960 | - | Entrepreneurship Field |
| [92] | 2003 | 755 | 1393 | 64 | 7 | 653 | 6 | Nascent Entrepreneurs |
| [83] | 2006 | 339 | 883 | 50 | 8 | 394 | - | Social Entrepreneurship |
| [85] | 2009 | 244 | 572 | 48 | 9 | 504 | - | Institutional Entrepreneurship |
| [93] | 1998 | 372 | 741 | 45 | 10 | 441 | - | Entrepreneurial Self-Efficacy |
| [72] | 1997 | 2602 | 3781 | 41 | - | 563 | 9 | Interfirm Networks |
| [94] | 1996 | 1419 | 2599 | 35 | - | 610 | 7 | Entrepreneurial Orientation |
| [95] | 2006 | 363 | 740 | 33 | - | 556 | 10 | Entrepreneurship Model |

The most downloaded/used document [89] in the double option representing the periods from 2013 and the last 180 days makes an attempt to specify the nature of abilities required to maintain a superior sustainable business performance in an open economy that is globally dispersed and characterised by rapid innovation. The second position is occupied by [78]. This is the document that showed the strongest increase in citations. Overall, the ranking also shows a certain thematic balance where new concepts associated with Social [83], or Institutional entrepreneurship [85] occupy a privileged position next to recurring and more usual topics like Opportunities [91], Models [95], Intention [80] and Orientation [94] in entrepreneurship.

## 5. Conclusions

Research in entrepreneurship has focused its attention on deciphering this topic contribution to social and economic development [96–99], and this idea has been what has attracted the greatest attention from academics and political institutions. Defining the discipline [3] and the figure of the entrepreneur is a task of large proportions, which is made even more complicated when taking into account the spectacular increase in literature. The efforts of a significant nucleus of authors [100–102] have led to more and more systematic and objective methods such as bibliometrics, which have helped decipher the theoretical and multidisciplinary framework that supports entrepreneurship and have been a starting point for new and more effective research agendas.

This analysis shows that innovation has historically been the vertebral axis of the field. However, the relationship between innovation and entrepreneurship has been explored very little as suggested by [29], and despite their common roots [4,103], there are few overlaps between the two knowledge platforms [104,105]. In that sense, debates are opening that call for a deeper analysis of the mentioned relationship. For example, [33] points to the innovating entrepreneur as the real architect of economic growth and questions the wisdom of the majority of politicians around the world who dedicate enormous amounts of money to finance self-employed workers suffering from low growth, a lack of resources and showing hardly any or no interest at all in innovation.

On the other hand, the analysis also shows how the concept of entrepreneur/entrepreneurship has changed and adapted itself to our current society, expanding into specific areas such as sustainable, social, institutional or other specific types of entrepreneurship. This new concept also demands a more in-depth examination of entrepreneurship's relationship with innovation, in line with arguments like those dealt with in [12,106].

In 2010, [39] pointed out how entrepreneurship research had become a huge "melting pot" of concepts and theories of many different disciplines and exposed how academics face the enormous challenge of building more and more systematic investigations with greater theoretical support. The article presented contributes to providing an objective and replicable understanding of the existing research on entrepreneurship and indicates emerging trends in this field of research. The results,

although they must be contrasted by expanding the search terms and the documentary sample, show how the fundamental focus of the discipline has been and currently is on innovation, although the concept of entrepreneur is increasingly broad and inclusive. Therefore, progress in clarifying this relationship is one of the main priorities for the progress of the discipline.

### 5.1. Main Findings and Future Perspectives

The co-word analysis made it possible to structure the data on various analysis levels, such as link networks or nodes, interactive network distributions and transformation of cognitive networks over various periods [28]. The obtained structures provided a focus to help track the thematic evolution of a representative sample of entrepreneurship-related documents with high citation rates (1968–2016). Main cognitive lines are:

- Performance/Innovation/Competitive Advantage;
- Entrepreneurial Firm/Market/Innovation/Strategy;
- Corporate Entrepreneurship/Management/Innovation/Market;
- Individual Trait/Self-Employment;
- Market Imperfections/Industry;
- Managers/Evolution/Performance/Innovation;
- Discovery/Field.

The study finds one of the centres of attention of entrepreneurship research to be its relationship with innovation. There is still no clear line separating both concepts and most affirmations made for innovation can be also applied to entrepreneurship, as "in the literature of field innovations, it has always been presented as the most important factor to achieve both economic and employment growth" [107] (p. 251). Scientists should explore this relationship in depth, both in qualitative and quantitative terms. The words innovation and entrepreneurship are often used indifferently even though they are not the same. Only some companies and entrepreneurs are innovators in the sense that they produce new products or services for customers and competitors. Even fewer are those that use radically innovative products and technologies [108,109].

Furthermore, the potential of bibliometrics to progress in the understanding of the discipline has been proven. There is little literature dealing with research trend evaluations, but there are new tools and metrics to help identify such trends. A bibliometric analysis should not be limited to knowing the number of citations or downloads for a specific document at a particular moment. If the aim is to detect trends, their progression must also be observed. The analysis of the evolution of citations and the usage counts have shown how the concept of entrepreneur has widened in response to increasing global challenges, which require an approach to new issues such as inequality, national vulnerabilities, financial crises, natural disasters or climate change. As a result, the evolution of citations and downloads points towards a closer association with ideas of Social Entrepreneurship, Sustainable Business and New Business Models.

### 5.2. Future Research

This study has revealed a series of questions that deserve closer attention in the future. The relationship between entrepreneurship and innovation needs to be clarified from a theoretical as well as a practical point of view, and more research is needed to investigate the role of innovation in the new and more specific concept of entrepreneur. Do all entrepreneurs necessarily need to be innovators? What is the impact that innovation and entrepreneurship have on new social contexts? Under which circumstances does innovation occur in entrepreneurship? Should political and institutional actors support all kinds of entrepreneurship?

From a bibliometric point of view, current analyses, which offer insight into our current understanding of the discipline, ought to be improved. More analyses are needed which take an in-depth look at the actual content of the documents. They are the texts that contain all the most

important conclusions and paradigms that make up the discipline. This requires the collaboration between scientists of both fields and the construction of specific indicators for entrepreneurship as well as the application of new advances in bibliometrics and other new metrics.

### 5.3. Limitations

This work is not exempt of limitations. Firstly, it can be questioned how representative the documents that constitute the sample actually are, since only a single search term, "entrepr*", was used to compile them. This type of strategy had been used previously in bibliometric studies in entrepreneurship [44,60,110] and represents a controversial decision, as it excludes documents related to entrepreneurship but dealing specifically with "Intrapreneurship", "Small Firms", "Small Enterprises", "Entry Firms", etc. However, given the multidisciplinary character of entrepreneurship and the fact that this root appeared in 108 document titles, the decision was made to continue with its use as no standard criteria exist in literature. Despite it is logical to think that the root used to obtain the set of documents (entrepr*) is the one with the highest frequency of appearance in the literature on entrepreneurship, there are authors who contribute to the debate and do not use this root or do it with a different meaning [60]. In this sense, although the findings obtained are significant, they are related to a concrete sample of 205 highly cited documents. Therefore, the analysis can be considered as a case study that has to be contrasted by expanding the list of terms applying a more efficient method to obtain lists of keywords that can represent a research domain [111].

The second limitation relates to the actual methodology used. Co-word analyses depend specifically on words, and those can appear in different forms and with different meanings [46]. Moreover, even though the normalisation process and keyword inclusion were exhaustive, the possibility of errors does exist.

Thirdly, the trend analysis that was performed depends on citations and usage counts. These are indicators with their own limitations [112–114]. Not all of them measure impact and not all scientists are able to download all documents. On the other hand, there are different reasons for citing and downloading documents that are not directly related to the actual influence on the research, sometimes articles are cited or downloaded due to the relevance of their author or other reasons and there are important factors differentiating between citations and downloads. Furthermore, it was decided to explore the evolution of citations after a period of two and a half years had elapsed. This choice is clearly arbitrary although it was deemed sufficiently long to observe a significant evolution.

Finally, it was also a limitation to use a single database (Web of Science) to obtain the sample. There are comparisons between this and other databases such as Google Scholar and Scopus [115,116], and the proposed methodology required a single pattern of citation, which prevented combining the three databases that would have been the best solution to obtain the greatest documentary coverage. However, it was finally decided to use WoS, since it is a complete database, it has the most influential journals in entrepreneurship and it has a well-established academic strength [117].

Taking into account all these limitations, this study could represent an initial step in trend analysis research in entrepreneurship, using an alternative focus which examines both the evolution of citations as well as the data provided by new metrics that have previously not been considered. Clearly, the analysis and the resulting thematic evolution should not be taken as a simple or definitive answer but serve to illustrate the potential of this type of study in a discipline that needs more conceptual and theoretical research.

**Supplementary Materials:** The following is available online at http://www.mdpi.com/2071-1050/11/11/3121/s1, Table S1: Sample of documents obtained with H-Classic methodology.

**Author Contributions:** The research is designed and performed by L.J.C.R. The data was collected by L.J.C.R. and F.J.F.-G. Analysis of data was performed by L.J.C.R. and S.M.S.-C. Finally, the paper is written by L.J.C.R., S.M.S.-C. and F.J.F.-G. All the authors read and approved the final manuscript.

**Acknowledgments:** The authors would like thank Hans Landström for his bibliometric research on entrepreneurship.

**Conflicts of Interest:** The authors declare no conflict of interest.

# Appendix A

**Table A1.** Studies related to bibliometric analysis about Entrepreneurship as a global field.

| Nº | Title | Authors | Publication Year | Total Citations Web of Science | Total Citations Scopus | Total Citations Google Scholar |
|---|---|---|---|---|---|---|
| 1 | Social structuration of the field of entrepreneurship: A case study | Dery, R, Toulouse, JM | 1996 | 18 | 18 | 48 |
| 2 | Who is publishing the entrepreneurship research? | Shane, SA | 1997 | 47 | - | 155 |
| 3 | Identifying current trends in entrepreneurship research: A new approach | Reader, D, Watkins, D | 2002 | - | - | 7 |
| 4 | Entrepreneurship research in emergence: Past trends and future directions | Busenitz, LW, West, GP, Shepherd, D, Nelson, T, Chandler, GN, Zacharakis, A | 2003 | 388 | 417 | 1220 |
| 5 | Intellectual structure of entrepreneurship research: A bibliometric study, 1956–2003 | Ramos, R.A. | 2004 | - | - | 0 |
| 6 | The field of entrepreneurship: a bibliometric assessment | Schildt, HA, Sillanpaa, A | 2004 | - | - | 16 |
| 7 | Entrepreneurial studies: The dynamic research front of a developing social science | Cornelius, B, Persson, O, Landstrom, H | 2006 | 87 | 96 | 283 |
| 8 | Is there conceptual convergence in entrepreneurship research? A co-citation analysis of Frontiers of Entrepreneurship Research, 1981–2004 | Gregoire, DA, Noel, MX, Dery, R, Bechard, JP | 2006 | 65 | 86 | 251 |
| 9 | Scholarly communities in entrepreneurship research: A co-citation analysis | Schildt, HA, Zahra, SA, Sillanpaa, A | 2006 | 100 | 114 | 243 |
| 10 | The social and collaborative nature of entrepreneurship scholarship: A co-citation and perceptual analysis | Reader, D, Watkins, D | 2006 | 42 | 50 | 104 |
| 11 | The past, present, and future of entrepreneurship research: Data analytic trends and training | Dean, MA, Shook, CL, Payne, GT | 2007 | 48 | 51 | 108 |
| 12 | Searching for" invisible colleges" in the Entrepreneurship literature | Ferreira, E.M. | 2009 | - | - | 0 |
| 13 | The entrepreneur, the organization and the world out there: A bibliometric review of 1239 papers on networks, social capital, cooperation, inter-organizational relations, and alliances in entrepreneurship | Sassmannshausen, S. P. | 2009 | - | - | 2 |
| 14 | The evolution of the literature on entrepreneurship. Uncovering some under researched themes | Teixeira, AAC, Santos, C. | 2009 | - | - | 12 |
| 15 | Entrepreneurship research: research communities and knowledge platforms | Landstrom, H, Persson, O | 2010 | 5 | 9 | 21 |
| 16 | Mapping the visible college(s) in the field of entrepreneurship | Teixeira, AAC | 2011 | 35 | 39 | 83 |

**Table A1.** *Cont.*

| Nº | Title | Authors | Publication Year | Total Citations Web of Science | Total Citations Scopus | Total Citations Google Scholar |
|---|---|---|---|---|---|---|
| 17 | Perspective Research Entrepreneurship Output Performance in 1992–2009 | Chen, JKC, Ho, YS, Wang, MH, Wu, YR | 2011 | 0 | 0 | 3 |
| 18 | The Intellectual Influence of Entrepreneurship Journals: A Network Analysis | Dos Santos, BL, Holsapple, CW, Ye, Q | 2011 | 8 | 9 | 16 |
| 19 | Charting the Growth of Entrepreneurship: A Citation Analysis of FER Content, 1981–2008 | Kushkowski, J.D | 2012 | - | 3 | 7 |
| 20 | Entrepreneurship: Exploring the knowledge base | Landstrom, H, Harirchi, G, Astrom, F | 2012 | 88 | 104 | 321 |
| 21 | Mapping the Intellectual Structure of Entrepreneurship Research: revisiting the invisible college | Campos, HM, Parellada, FS, Palma, Y | 2012 | 3 | 5 | 18 |
| 22 | Intellectual structure of the entrepreneurship field: a tale based on three core journals | Teixeira, AAC, Ferreira, E.M. | 2013 | - | - | 3 |
| 23 | A visual analytic study of articles in entrepreneurship research | Yu, L. -C, Tang, T.-I | 2014 | - | 0 | - |
| 24 | Computational and visual analysis of the development stage of theories in the social sciences: a case in the entrepreneurship field | Qian, G | 2014 | 0 | 0 | 0 |
| 25 | Entrepreneurship Research (1985–2009) and the Emergence of Opportunities | Busenitz, LW, Plummer, LA, Klotz, AC, Shahzad, A, Rhoads, K | 2014 | 25 | 36 | 112 |
| 26 | Origin and emergence of entrepreneurship as a research field | Meyer, M, Libaers, D, Thijs, B, Grant, K, Glanzel, W, Debackere, K | 2014 | 15 | 23 | 56 |
| 27 | Trends in and contributions to entrepreneurship research: a broad review of literature from 1996 to June 2012 | Luor, TY, Lu, HP, Yu, HJ, Chang, KL | 2014 | 7 | 10 | 24 |
| 28 | Entrepreneurship across regions: Internationalization and/or contextualization? | Landstrom, H, Jing, S, Quinghua, Z. | 2015 | - | 0 | 3 |
| 29 | Entrepreneurship Research Dynamics (1992–2013): Aim at Entrepreneurial, Innovative Firms and Business Operations | Chen, JKC | 2015 | 0 | 0 | 0 |
| 30 | Entrepreneurship research in three regions-the USA, Europe and China | Landstrom, H, Jing, S, Zhai, QH | 2015 | 2 | 5 | 15 |
| 31 | The evolution of the small business and entrepreneurship field: A bibliometric investigation of articles published in the International Small Business Journal | Volery, T, Mazzarol, T | 2015 | 6 | 7 | 23 |
| 32 | Thirty years of entrepreneurship research published in top journals: analysis of citations, co-citations and themes | Ferreira, M.P., Reis, N.R., Miranda, R. | 2015 | - | - | 26 |
| 33 | Entrepreneurship and Family Firm Research: A Bibliometric Analysis of An Emerging Field | Lopez-Fernandez, MC, Serrano-Bedia, AM, Perez-Perez, M | 2016 | 11 | 9 | 25 |

**Table A1.** *Cont.*

| Nº | Title | Authors | Publication Year | Total Citations Web of Science | Total Citations Scopus | Total Citations Google Scholar |
|---|---|---|---|---|---|---|
| 34 | Entrepreneurship as a dynamic field of study: a bibliometric analysis of research output | Cabeza-Ramirez, LJ, Canizares, SMS, Fuentes-Garcia, FJ | 2017 | - | - | 1 |
| 35 | Entrepreneurship research: mapping intellectual structures and research trends | Ferreira, J.J.M., Fernandes, C.I., Kraus, S. | 2017 | - | 4 | 6 |
| 36 | Characterisation of the classics of entrepreneurship (1968–2016). An analysis based on Web of Science | Cabeza-Ramirez, LJ, Canizares, SMS, Fuentes-Garcia, FJ | 2018 | 0 | 0 | 0 |
| 37 | Contributing Forces in Entrepreneurship Research: A Global Citation Analysis | Xu, NH, Chen, YN, Fung, AN, Chan, KC | 2018 | 0 | 0 | 1 |
| 38 | Entrepreneurship and regional development. A bibliometric analysis | Dan, MC, Goia, SI | 2018 | 0 | 0 | 0 |
| 39 | Mapping the evolution of entrepreneurship as a field of research (1990–2013): A scientometric analysis | Chandra, Y | 2018 | 0 | 2 | 6 |
| 40 | The social structure of entrepreneurship as a scientific field | Landstrom, H, Harirchi, G | 2018 | 0 | 1 | 3 |

Citations collected on the 7 October 2018 in Scopus and WoS. The document sample was taken from a systematic literature review which is still pending publication. More details regarding document selection in [56]. Search keywords: root Entrepr* and: bibliometric, infometric, webometric, citation analysis/citation, analyses, direct citation/direct citations, cocitation analysis/cocitation analyses/co-citation analysis/co-citation analyses, bibliographic coupling, coword analysis/coword analyses/co-word analysis/co-word analyses, coauthorship/coauthorship network/coauthorship networks/co-authorship network/co-authorship networks, self citation/self citations/self-citation/self-citations, network analysis/networks analyses (refine by bibliometric), citation map, citation visuali*, science policy (refine by bibliometric), research policy (refine by bibliometric), impact factor/impact factors (refine by bibliometric), h-index/h index/hirsch index, patent analysis/patent analyses (refine by bibliometric), zipf, Bradford, lotka, Intellectual structure (refine by bibliometric).

**Table A2.** Bibliometric Studies about Entrepreneurship specific areas.

| Nº | Title | Authors | Publication Year | Total Citations WoS |
|---|---|---|---|---|
| 1 | Family firms - On the state-of-the-art of business research | Harms, R, Kraus, S, Filser, M, Gotzen, T | 2011 | 7 |
| 2 | State-of-the-art current research in international entrepreneurship: A citation analysis | Kraus, S | 2011 | 19 |
| 3 | The bibliometric structure of spin-off literature | Wallin, MW | 2012 | 15 |
| 4 | A review of entrepreneurship education research through bibliometric perspective | Yu, LC, Yang, JM | 2013 | 0 |
| 5 | Social Entrepreneurship: An exploratory citation analysis | Kraus, S, Filser, M, O'Dwyer, M, Shaw, E | 2014 | 25 |
| 6 | Focus on China: the current status of entrepreneurship research in China | Su, J, Zhai, QH, Ye, MH | 2014 | 8 |
| 7 | Bibliographic analysis and strategic management research in Africa | Zoogah, DB, Rigg, JS | 2014 | 0 |
| 8 | A bibliometric study on the entrepreneurial orientation (2001–2013) | Saidi, S, Chebbi, H, Sellami, M, Weber, Y | 2014 | 0 |

**Table A2.** *Cont.*

| Nº | Title | Authors | Publication Year | Total Citations WoS |
|---|---|---|---|---|
| 9 | Structuring the Technology Entrepreneurship publication landscape: Making sense out of chaos | Ratinho, T, Harms, R, Walsh, S | 2015 | 14 |
| 10 | Who wants to live forever: exploring 30 years of research on business longevity | Riviezzo, A, Skippari, M, Garofano, A | 2015 | 4 |
| 11 | Entrepreneurship research in China: internationalization or contextualization? | Su, J, Zhai, QH, Landstrom, H | 2015 | 8 |
| 12 | The emergence of the knowledge spillover theory of entrepreneurship | Ghio, N, Guerini, M, Lehmann, EE, Rossi-Lamastra, C | 2015 | 48 |
| 13 | A co-citation bibliometric analysis of strategic management research | Ferreira, JJM, Fernandes, CI, Ratten, V | 2016 | 11 |
| 14 | What do we [not] know about technology entrepreneurship research? | Ferreira, JJM, Ferreira, FAF, Fernandes, CIMAS, Jalali, MS, Raposo, ML, Marques, CS | 2016 | 8 |
| 15 | The Phenomenon of Social Enterprises: Are We Keeping Watch on This Cultural Practice? | Goncalves, CP, Carrara, K, Schmittel, RM | 2016 | 3 |
| 16 | Is international entrepreneurship a field? A bibliometric analysis of the literature (1989–2015) | Servantie, V, Cabrol, M, Guieu, G, Boissin, JP | 2016 | 9 |
| 17 | A bibliometric analysis of social entrepreneurship | Rey-Marti, A, Ribeiro-Soriano, D, Palacios-Marques, D | 2016 | 17 |
| 18 | A bibliometric analysis of international impact of business incubators | Ribeiro-Soriano, D, Albort-Morant, G | 2016 | 14 |
| 19 | Some Predictors of Entrepreneurship Article Impact | Perry, JT, Hanke, RH, Chandler, GN, Markova, G | 2016 | 0 |
| 20 | University students' entrepreneurial intentions: A bibliometric study | Arias, AV, Restrepo, IM, Restrepo, AM | 2016 | 2 |
| 21 | Research on entrepreneurial orientation: current status and future agenda | Martens, CDP, Lacerda, FM, Belfort, AC, de Freitas, HMR | 2016 | 11 |
| 22 | Scientific production in the field of academic spin-off: A bibliometric analysis | Segui-Mas, E, Sarrion-Vines, F, Tormo-Carbo, G, Oltra, V | 2016 | 2 |
| 23 | Twenty Years of Rural Entrepreneurship: A Bibliometric Survey | Pato, ML, Teixeira, AA | 2016 | 11 |
| 24 | Analyzing informal entrepreneurship: a bibliometric survey | Ferreira, JJ, Dos Santos, EMMN | 2017 | 0 |
| 25 | Mapping the Intellectual Structure of Research on 'Born Global' Firms and INVs: A Citation/Co-citation Analysis | Garcia-Lillo, F, Claver-Cortes, E, Marco-Lajara, B, Ubeda-Garcia, M | 2017 | 2 |
| 26 | Innovation and entrepreneurship in the academic setting: a systematic literature review | Schmitz, A, Urbano, D, Dandolini, GA, de Souza, JA, Guerrero, M | 2017 | 12 |
| 27 | Let the best story win - evaluation of the most cited business history articles | Ojala, J, Eloranta, J, Ojala, A, Valtonen, H | 2017 | 1 |

**Table A2.** *Cont.*

| Nº | Title | Authors | Publication Year | Total Citations WoS |
|---|---|---|---|---|
| 28 | Modeling complex entrepreneurial processes A bibliometric method for designing agent-based simulation models | Shim, J, Bliemel, M, Choi, M | 2017 | 1 |
| 29 | Entrepreneurial university: towards a better understanding of past trends and future directions | Santos, G, Marques, CS, Mascarenhas, C, Galvao, AR | 2017 | 2 |
| 30 | Unpacking the innovation ecosystem construct: Evolution, gaps and trends | Gomes, LAD, Facin, ALF, Salerno, MS, Ikenami, RK | 2018 | 5 |
| 31 | A Systematic Review of International Entrepreneurship Special Issue Articles | Perenyi, A, Losoncz, M | 2018 | 0 |
| 32 | Entrepreneurial education: a bibliometric study on recent scientific production | Johan, DA, Kruger, C, Minello, IF | 2018 | 0 |
| 33 | Entrepreneurs' Well-Being: A Bibliometric Review | Sanchez-Garcia, JC, Vargas-Morua, G, Hernandez-Sanchez, BR | 2018 | 0 |
| 34 | A Research Agenda on Open Innovation and Entrepreneurship: A Co-Word Analysis | Mora-Valentin, EM, Ortiz-de-Urbina-Criado, M, Najera-Sanchez, JJ | 2018 | 0 |
| 35 | Entrepreneurial cognition and socially situated approach: a systematic and bibliometric analysis | Sassetti, S, Marzi, G, Cavaliere, V, Ciappei, C | 2018 | 0 |
| 36 | The Development of Sustainable Entrepreneurship Research Field | Sarango-Lalangui, P, Santos, JLS, Hormiga, E | 2018 | 0 |
| 37 | Where do we go from now? Research framework for social entrepreneurship | Macke, J, Sarate, JAR, Domeneghini, J, da Silva, KA | 2018 | 2 |
| 38 | Entrepreneurship and regional development. A bibliometric analysis | Dan, MC, Goia, SI | 2018 | 0 |
| 39 | Inspecting the Achilles heel: a quantitative analysis of 50 years of family business definitions | Hernandez-Linares, R, Sarkar, S, Cobo, MJ | 2018 | 2 |
| 40 | A look back over the past 40 years of female entrepreneurship: mapping knowledge networks | Santos, G, Marques, CS, Ferreira, JJ | 2018 | 0 |
| 41 | A bibliometric analysis of born global firms | Dzikowski, P | 2018 | 1 |
| 42 | Entrepreneurship and entrepreneurial ecosystems | Malecki, EJ | 2018 | 4 |
| 43 | Entrepreneurship education. A pathway to improve entrepreneurship orientation of the students | Iturralde, T, Maseda, A | 2018 | 0 |
| 44 | Bridging past and present entrepreneurial marketing research: A co-citation and bibliographic coupling analysis | Most, F, Conejo, FJ, Cunningham, LF | 2018 | 0 |
| 45 | Entrepreneurship education and training as facilitators of regional development A systematic literature review | Ferreira, JJ, Galvao, A, Marques, C | 2018 | 0 |

Citations collected on the 9 December 2018. Document search performed in the Web of Science Core Collection. Search sequence: topic (entrepreneurship)/refined by topic (bibliometric)/ timespan (all years)/Indexes (social citation index-expanded, social sciences citation index, a&hci, cpci-s, cpci-ssh, bkci-s.blco-ssh, esci, ccr-expanded, ic. Documents that did not comply with the condition of being a bibliometric analysis of a specific area in entrepreneurship were manually deleted from the list.

**Table A3.** Search file.

| WOS | |
|---|---|
| Date | 1 June 2016 |
| Place | Web of Science Core Collection |
| Search Type | Simple |
| Syntax | entrep*/topic |
| Filtered by area | (management or business or economics or planning development or history or social issues or education educational research or operations research management science or business finance or environmental studies or sociology or geography or political science or social sciences interdisciplinary or engineering industrial or history of social sciences or area studies or urban studies or public administration or computer science information systems or information science library science or psychology applied or international relations or multidisciplinary sciences or anthropology or psychology multidisciplinary or law) |

**Table A4.** Top 5 Documents by citations received in sample of 205 (H-Classics).

| Order | Title | Author | Year | Cites | % |
|---|---|---|---|---|---|
| 1 | Social structure and competition in interfirm networks: The paradox of embeddedness | Uzzi, B. | 1997 | 2602 | 3.3% |
| 2 | The promise of entrepreneurship as a field of research | Shane, S.; Venkataraman, S | 2000 | 2090 | 2.7% |
| 3 | Clarifying the entrepreneurial orientation construct and linking it to performance | Lumpkin, G.T.; Dess, G. G. | 1996 | 1419 | 1.8% |
| 4 | Explicating dynamic capabilities: The nature and microfundations of (sustainable) enterprise performance | Teece, D.J. | 2007 | 1124 | 1.4% |
| 5 | Market orientation and the learning organization | Slater, S.F.; Narver, J.C. | 1995 | 1115 | 1.4% |

Cites collected in 2016

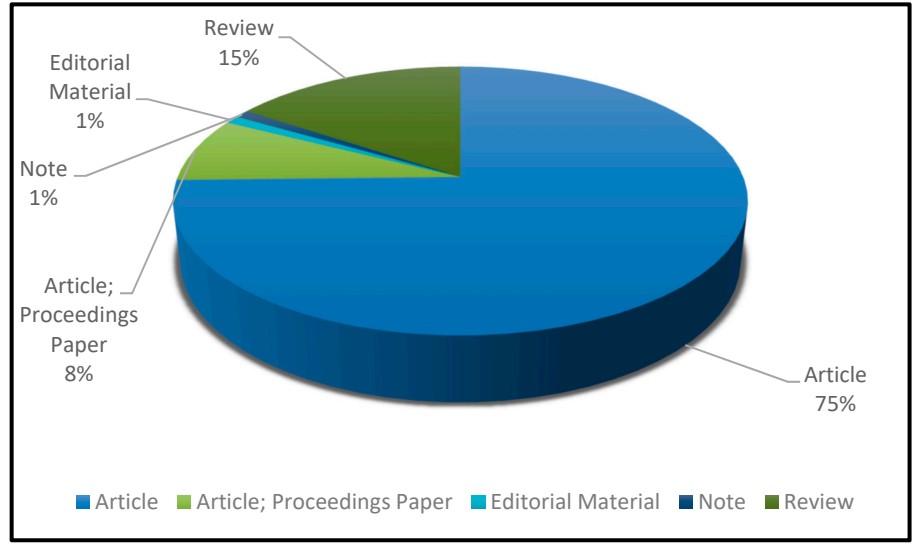

**Figure A1.** Document typology in Sample (H-Classic).

**Table A5.** Top 5 most visible H-Classic authors by number of citations.

| R. | Author | Number of Docs | Social Science Citation Index Cites | SSCI % | Affiliation | Country | Field of Research |
|---|---|---|---|---|---|---|---|
| 1 | Shane, S. | 7 | 4844 | 6.15% | University of Maryland College Park | USA | Economic/ Entrepreneurship |
| 2 | Uzzi, B. | 1 | 2602 | 3.3% | Northwestern University | USA | Sociology/Business Administration |
| 3 | Lumpking, G.T. | 4 | 2335 | 2.96% | University of Texas Arlington | USA | Entrepreneurship |
| 4 | Venkataraman, S. | 1 | 2090 | 2.65% | University of Virginia | USA | Business Administration |
| 5 | Dess, G. G. | 3 | 2077 | 2.64% | Arizona State University | USA | Management |

**Table A6.** Top 5 most cited authors in H-Classic (Knowledge Base).

| Ranking | Author | References | Documents |
|---|---|---|---|
| 1 | Schumpeter J. A. | 7 | 71 |
| 2 | Aldrich H.E. | 38 | 54 |
| 3 | Shane S. A. | 29 | 46 |
| 4 | Barney J. B. | 23 | 44 |
| 5 | Porter M. E. | 20 | 44 |

**Table A7.** Top most influential documents among the H-Classics (Knowledge Base).

| R. | Author | Title | Year | Number of Documents |
|---|---|---|---|---|
| 1 | Schumpeter, J.A | The theory of economic development | 1934 | 53 * |
| 2 | Barney, J.B. | Firm resources and sustained competitive advantage | 1991 | 36 |
| 3 | Shane S. y Venkataraman S. | The promise of entrepreneurship as a field of research | 2000 | 30 |
| 4 | Lumpkin, G. T. & Dess | Clarifying the entrepreneurial orientation construct and linking it to performance | 1996 | 27 |
| 5 | Schumpeter, J.A | Capitalism, socialism and democracy. | 1942 | 26 * |

* The same reference can appear in several documents.

**Table A8.** Top 5 Journals H-Classic.

| R. | Journal | Number of Docs | SSCI Citations | SSCI% |
|---|---|---|---|---|
| 1 | Journal of Business Venturing | 24 | 8338 | 11% |
| 2 | Strategic Management Journal | 23 | 9652 | 12% |
| 3 | Academy of Management Journal | 16 | 4994 | 6% |
| 4 | Academy of Management Review | 12 | 7839 | 10% |
| 5 | Administrative Science Quarterly | 9 | 6448 | 8% |

**Table A9.** Top 5 Journals Knowledge Base (references in H-Classic).

| R. | Journal | References | % |
|----|---------|-----------|---|
| 1 | Journal of Business Venturing | 239 | 2.54% |
| 2 | Administrative Science Quarterly | 200 | 2.12% |
| 3 | Academy of Management Review | 199 | 2.11% |
| 4 | Academy of Management Journal | 256 | 2.72% |
| 5 | Strategic Management Journal | 321 | 3.41% |

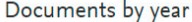

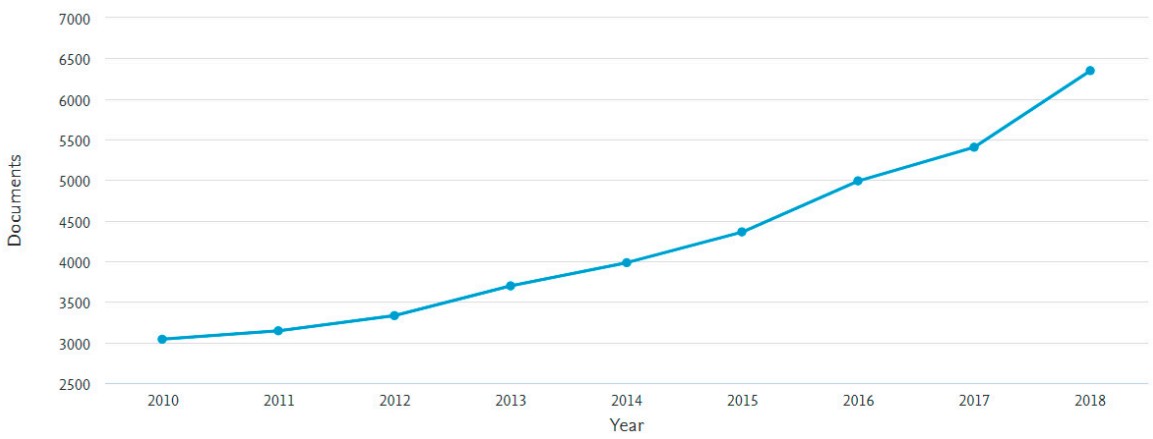

**Figure A2.** Number of papers about entrepreneurship. Source Scopus (Title-Abs-Key (entrepreneurship) or Title-Abs-Key (entrepreneur) and limit-to Pubyear 2010 to 2018.

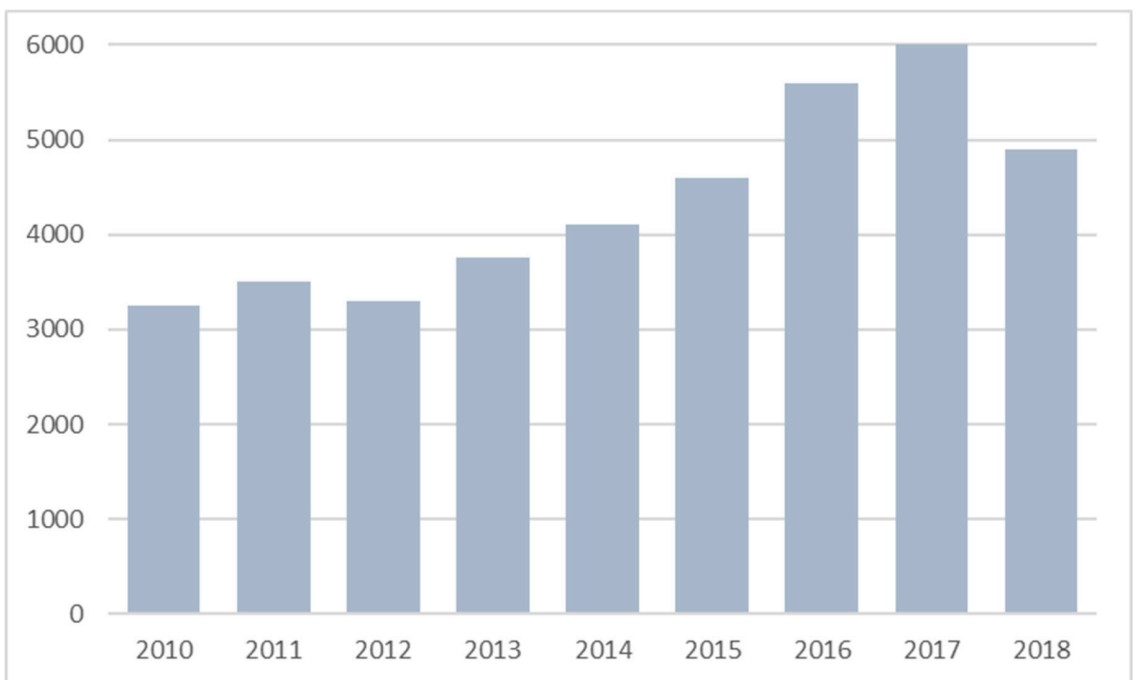

**Figure A3.** Number of papers about entrepreneurship. Source WoS; Searched for Topic: (entrepreneurship) or topic (entrepreneur); refined by publication year 2010 to 2018.

**Table A10.** Algorithm configuration of simple centres.

| Settings | Origins Period 1 (1968–1995) | Development 1 Period 2 (1996–2000) | Development 2 Period 3 (2001–2005) | Consolidation Period 4 (2006–2011) |
|---|---|---|---|---|
| Number of Documents | 45 | 54 | 86 | 20 |
| Min-occurrences | 2 | 2 | 3 | 2 |
| Min-Co-occurrences | 2 | 2 | 3 | 2 |
| Min-Keywords | 2 | 2 | 2 | 2 |
| Max-Keywords | 5 | 5 | 5 | 5 |

The optimal configuration must be sufficiently balanced to avoid exceedingly high values that form few links or excessively low that forms unintelligible networks without bibliometric sense. It was made according to the total number of documents that form the sample (205), the temporary subdivision (four periods), the number of documents that make up each period and the words associated with each period. For this reason, the requirements of period 3 were made stricter.

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
