# Peer review of "Past Themes and Tracking Research Trends in Entrepreneurship: A Co-Word, Cites and Usage Count Analysis"

_sustainability, doi:10.3390/su11113121_

Round 1

Reviewer 1 Report

Title (lines 1-3): No reason to reuse the word entrepreneurship twice. Re-word the title.

Paragraphs: The paper has to many paragraphs. Re-structure. 

Lines 36-37: needs a reference.

Line 39-42: break long sentence in two.

Line 41: remove capital from Entrepreneurship.

Line 48 and 49: Final part of sentence does not make sense. Remove or rephrase in a new sentence.

Line 59 and 61: Generally, when referring to a table or figure it should be capitalized e.g. Table 1 or Figure 1. Fix throughout unless journal guidelines differ.

Line 59: Are you specifically referring to the discipline of entrepreneurship? This is also not clearly pointed out in Table 1. What keywords did you use to search the data in Table 1?

Lines 62-65: This is not clear? You mention that “a search carried out in different databases clearly shows that few works of this type….. “  Firstly, which databases’ are you referring to and secondly, when one adds the keyword entrepreneurship thousands of published articles appear. The rest of the statement is also unclear. Where did you read that 3000 document is written each year and are you referring to the phenomenon of entrepreneurship (not clear). Which analysis are you reefing to in Line 64? What content and ideas are you referring to in line 65.

Refer to the following paper:

Ahl H.J., 2002, The making of the female entrepreneur: A discourse analysis of research texts on women’s entrepreneurship, Jönköping Sweden: Parajett AB.

De Bruin A., Brush A.G., Welter F., 2006, Introduction to the special issue: Towards building cumulative knowledge on women’s entrepreneurship, “Entrepreneurship Theory and Practice”, September Issue, 30(5).

De Bruin A., Brush C.G., Welter F., 2007, Advancing a framework for coherent research on women’s entrepreneurship, “Entrepreneurship Theory and Practice”, May Issue, 31(3).

Meyer, N. 2018. Research on female entrepreneurship: Are we doing enough? Polish Journal of Management Studies, 17(2):158-169.

Line 67-68: These types of content analysis has been done, so maybe not so rarely? (see above sources).

Line 108: fix…. in deepening their understanding nor in deepen in their understanding.

Line 120-121: Are you referring only to WoS publication?

Line 122: Are the 7500 documents only related to entrepreneurship. Be more specific.

Line 162: Rephrase sentence.

Line 169-181: Although I agree with the statement “New research methods tend to require the most influential articles to be downloaded” in some cases researchers may tend to download papers of well-known authors in the field (e.g. Schumpeter, Thurik, Wennekers, Shane, Bird, Brush, Venkataraman, Audretsch, Lumpkin etc.) and not necessarily the newest and latest “top studies”

Line 187-192: I find this somewhat problematic to refer to the methodology used in another study and not discuss it in this study. Explain the methodology referred [50 and 51] in this study as well.

Line 201: Write out the abbreviation SSCI if used for the first time.

Line 201: Why was only 205 papers retrieved and who/which one classified ad the so-called “classics”

Line 254: Refer to note Line 187-192.

Line 256: Again, who are these classic authors and how were they selected?

Line 260: Explain the H-Classics method.

Figures 4-7: If the document is printed, the text almost disappears. Make sure the text is clear enough for electronic and monotone print version.

Line 268 and 278: you refer to 45 document from period 1 although in Table 1 there is only a total of 34 (N. of documents).

Line 270, 272, 300, 338 and 354: Same as previous line. Second time period 54 but in Table only 43; 3rd period 86 vs 74 in Table 1 and 20 in 4th period vs 21in Table 1.

Lines 324-334: Delete

Figure 8 - 9: Very complex figure with many of the headings/labels not readable.

Tables 1 and 2: The use of . and , in the numerical values is confusing. I suggest drop the . in the sum of citations column (e.g. 21 949 instead of 21.949). And also, is the average citations for example 342 953 or 342,953. Simplify it as it is confusing at the moment.

Table 2: The number of documents (N. of documents) adds up to 229 not 205 (as sample) or 172 (total of docs in Table 1). Please clarify.

Line 422: You mention that line 6 and 7 show the greatest activity. Explain?

Line 433: Which four themes? Are you referring to social-value, success, orientation and joint ventures? If yes, make this clearer.

Line 461 and 463: Perhaps round off the percentage e.g. 218.36% to 218%

Table 3: Again, round of the %

Table 3 and 4: Is the Document and Doc. the number of the document (ID) or the quantity? Clarify. Perhaps say Doc Nr. or Doc ID in the heading.

Line 493 to 500: Document [77], is this referring to the citation in reference list [77] or the doc nr/ID from Tables 3 and 4? This is confusing to reader.

Line 512: Should Innovation be capitalized?

Line 523: See comments line 512

Line 532. Restructure sentence not to start with a numerical 7

The paper needs a conclusion section

 This was a very interesting paper to read and I look forward to read the final published version. Congratulations on a good paper!

Author Response

Please find attached the document with the answers

Reviewer 2 Report

The authors use bibliometric content analysis to uncover developing trends in entrepreneurship in published research. An interesting approach, provides some insight into the development of the entrepreneurship discipline. The main points are as follows:

- on page 13 "thematic composition it can be concluded", provide further methodological evidence to support how these lines were constructed and how the conclusions were reached

- given the close association found between innovation and entrepreneurship, the search term entr* in article titles as a limitation needs to be expanded on in the Discussion where such associated is discussed, seems to indicate the search criteria is insufficient

- opening sentence "Entrepreneurship combines multiple realities, all of which are linked to the idea of actions that promote the creation of something new, different and valuable [1]." Narrow view and ignores many documented forms of entrepreneurship and the ongoing discourse on what constitutes entrepreneurship, the reference is inadequate, see your own statement on page 3 with citation [26]

- on page 2 sentences starting with "A search carried out...", meaning is completely obscure, clarify "works of this type" and the process of discover to support the claims

- on page 12 co-word analysis is conducted from one period to the next, suggest be shown across all periods

- remove statement "content analysis, rarely performed until now." unless can show proof, also suggest removing "first time" statements

Other general items are:

- explain how to read table 1 and figure 8, in particular the ','

- increase text size in figures

- on page 13 in bullet points references table 5 but there is no such table

- paragraph starting at line 260 be moved to be included in paragraph starting line 201

- grammatical item, avoid use of third party pronouns such as 'it' and 'they' as often cannot discern who or what is being referenced, using these pronouns makes reading and understanding less clear

- some incorrect words and spelling errors

Author Response

Please find attached a word file with the answers

Reviewer 3 Report

Dear Authors,

Thank you very much for the opportunity to read your paper focused on bibliometric analysis of trends in entrepreneurship research and its perception. I find the article very interesting as it offers exciting overview of what is mostly cited in the field and how the topics emerged from the year 2013 and on. At the same time, I would like to propose authors series of comments that might serve as an inspiration towards strengthening of this promising paper. At any point, I would like to wish the authors good luck in the development of their paper!

1.      I believe that the authors might reflect their observations in the continuous initiative of Kuckertz and Prochotta (2018) titled “What is hot in entrepreneurship research.”

2.      I think that the authors should correct several minor issues: Clarivate Analytics now administers the Web of Science database, the authors should also state the exact dates when they made the extraction of the data, and I believe that it would be very beneficial to add sources to other data sources into the reference list.

3.      I find quite unfortunate that the authors do not also talk about the sources of the data. Of course, we know what the key entrepreneurship journals are, but still, it might be very relevant to add at least basic information on the sources and frequencies/maps/graphs.

4.      In the introduction, the authors might reflect their perception of entrepreneurship in a holistic way. Nowadays we talk more and more about the entrepreneurship ecosystem (Stam, 2015) and one of the disciplines which should be definitely mentioned is the economics of entrepreneurship and self-employment (Dvouletý, 2018).

5.      In my opinion, the authors should take more time and efforts and think about the added value of the paper. In my opinion, the added value should also be communicated in the abstract and in the concluding sections in a better and extensive way.

6.      The authors definitely know that there are Scopus and Google Scholar databases that are very popular for searching for academic resources. I very much understand the reasons why they focused only on Web of Science, but still, this should be discussed at least as a limitation of the study.

List of References

Dvouletý, O. (2018). How to analyse determinants of entrepreneurship and self-employment at the country level? A methodological contribution. Journal of Business Venturing Insights9, 92-99.

Kuckertz, A., & Prochotta, A. (2018). What’s hot in entrepreneurship research 2018?. Online: http://opus.uni-hohenheim.de/volltexte/2018/1462/pdf/2018_Kuckertz_Prochotta_HERB.pdf

Stam, E. (2015). Entrepreneurial ecosystems and regional policy: a sympathetic critique. European Planning Studies23(9), 1759-1769.

Author Response

Please find attached a word file with the answers. Thank you.

Round 2

Reviewer 1 Report

Dear Authors

The comments were well incorporated into the final document and I hope it contributed to improving the study in your opinion. It is an interesting piece of work. 

I will look out for the final published version.

Regards

Author Response

Dear Reviewer,

Thank you very much for your words and the time you devoted to the evaluation of this paper. We really appreciate your suggestion and we strongly feel they have contributed to improve this work.

Regards

The authors

Reviewer 2 Report

The authors have made a significant improvement in the paper to address my comments as well as the comments of the other reviewers. The addition of expanded explanations of the methodology along with the added or revised figures and tables helped in increasing understanding and following the logic. The authors in places state their opinions as if these opinions were facts without providing any supporting references, such occurrences need to be reworded or appropriate references supplied.  The main revision still needed is a very thorough copy proof reading to correct grammar errors, inconsistent usage of words/phrases/acronyms, and readability of figures. Note: examples are provided to illustrate the type of problem and as such the list of examples is not exhaustive. The authors are responsible to find and correct all occurrences of these problems. 

Some specific examples are 

as mentioned in my previous comments, there is an excessive use of third party pronouns, the use of these words confuses the reader, see paragraph starting at line 52 which contains these pronouns 14 times

the paragraph at line 52 is also a good example of opinions, such as with the word 'extraordinary' and the phrase 'Alof them draw economic and personal satisfaction'

need to add page numbers

World of Science appears in various and inconsistent forms either in the expanded form or the acronym (Wos, WOS, WoS, woS), also the first occurrence of WoS is used before being defined

quotes without page numbers

some text in many of the figures so small cannot read, even if zoom in the text remains blurry, in figure 1 the box in the lower left corner

line 266, for readers unfamiliar with the term 'motor' when discussing themes provide a short definition

figure 4, 5, 6 - Strategic diagram; in each figure the circles represent different values from tables 1, 2, 3; make the figures consistent in what they represent and aligned with the text explanation

tables 1, 2, 3 - Summary of bibliometric indicators; right or decimal align the columns of numbers; in the column labeled 'Average  n. of citations' have all numbers display 2 decimal places

MDPI logo appears in the header on a few pages

in table 5 have a more lighter and translucent shade of blue

switch the order of paragraph at line 573 with paragraph at line 580

add a running heading to the tables in the appendix

order the years in graph 3 to be the same as in graph 2

reference 21 & 56 incomplete

Author Response

Dear Reviewer,

Thank you very much for your words and the time you devoted to the evaluation of this paper. We really appreciate your suggestions and we strongly feel they have contributed to improve this work.

These are the corrections we have made to attend your comments.

·         as mentioned in my previous comments, there is an excessive use of third party pronouns, the use of these words confuses the reader, see paragraph starting at line 52 which contains these pronouns 14 times

We have tried to substitute this type of pronouns when the sentences were not clear all along the document.

·         the paragraph at line 52 is also a good example of opinions, such as with the word 'extraordinary' and the phrase 'All of them draw economic and personal satisfaction'

We have added a reference or change the sentence when a support was needed.

·         need to add page numbers

Page number are in the upper right corner according to the Journal template. We have added the page number in the horizontal pages as they did not appear before.

·         World of Science appears in various and inconsistent forms either in the expanded form or the acronym (Wos, WOS, WoS, woS), also the first occurrence of WoS is used before being defined

It has been homogenized and it has been described in the first occurrence

·         quotes without page numbers

Page numbers have been added in references with quotes.

·         some text in many of the figures so small cannot read, even if zoom in the text remains blurry, in figure 1 the box in the lower left corner

We have increased the size for figure 1. We have checked all the figures and we are able to see the text in all of them, at least in our word and pdf files.

·         line 266, for readers unfamiliar with the term 'motor' when discussing themes provide a short definition

We have provided a short definition and a reference

·         figure 4, 5, 6 - Strategic diagram; in each figure the circles represent different values from tables 1, 2, 3; make the figures consistent in what they represent and aligned with the text explanation

We have homogenized the figures according to the tables. We have also added what the figures represent although it is also explained in the text.

·         tables 1, 2, 3 - Summary of bibliometric indicators; right or decimal align the columns of numbers; in the column labeled 'Average  n. of citations' have all numbers display 2 decimal places

We have homogenized these tables according to your comment

·         MDPI logo appears in the header on a few pages

We have solved this problem

·         in table 5 have a more lighter and translucent shade of blue

We have used a lighter blue color

·         switch the order of paragraph at line 573 with paragraph at line 580

We have switched the order between both paragraphs

·         add a running heading to the tables in the appendix

We have added the running heading for tables 1 and 2

·         order the years in graph 3 to be the same as in graph 2

We have re-ordered the years in graph 3

·         reference 21 & 56 incomplete

We have completed these references

Reviewer 3 Report

Dear Authors, 

thank you very much for incorporating my comments and suggestions into the paper. I believe that the current version of the paper is much stronger and thus, I have no objections about publishing the current version of the paper.

Your Reviewer

Author Response

Dear Reviewer,

Thank you very much for your words and the time you devoted to the evaluation of this paper. We really appreciate your suggestions and we strongly feel they have contributed to improve this work.

Regards

The authors